# Calibrated Probabilistic Forecasts for Arbitrary Sequences

**Charles Marx**                                              *ctmarx@cs.stanford.edu*
*Department of Computer Science*
*Stanford University*

**Volodymyr Kuleshov**                                        *kuleshov@cornell.edu*
*Department of Computer Science*
*Cornell Tech*

**Stefano Ermon**                                             *ermon@cs.stanford.edu*
*Department of Computer Science*
*Stanford University*

**Reviewed on OpenReview:** *https://openreview.net/forum?id=nuIUTHGlM5*

## Abstract

Real-world data streams can change unpredictably due to distribution shifts, feedback loops and adversarial actors, which challenges the validity of forecasts. We present a forecasting framework ensuring valid uncertainty estimates regardless of how data evolves. Leveraging the concept of Blackwell approachability from game theory, we introduce a forecasting framework that guarantees calibrated uncertainties for outcomes in any compact space (e.g., classification or bounded regression). We extend this framework to recalibrate existing forecasters, guaranteeing calibration without sacrificing predictive performance. We implement both general-purpose gradient-based algorithms and algorithms optimized for popular special cases of our framework. Empirically, our algorithms improve calibration and downstream decision-making for energy systems.

## 1 Introduction

Machine learning models are often used in settings where data changes unpredictably after deployment. For example, in medical diagnosis, epidemiological trends can induce *distribution shifts*, altering the prevalence of diseases in a population. Furthermore, a model's predictions can impact future observations through *feedback loops*–such as in financial forecasting (De Long et al., 1990), adaptive experimentation (Deshpande & Kuleshov, 2021), preventative medicine (Adam et al., 2020), and predictive policing (Ensign et al., 2018)–where the model's deployment induces nonstationarity in future data. Even more challenging, in settings such as security, regulation, and game playing, data may be chosen *adversarially* to invalidate predictions.

Accurate uncertainty estimates are crucial for informed decision-making in dynamic environments. This work addresses the challenge of maintaining forecast reliability despite adversarial or unpredictable data shifts. Take for example, an electrical grid operator tasked with using energy generation and load forecasts to ensure reliable energy supply. Uncertainty estimates are needed to *evaluate the risk* of different management decisions, allowing the operator to guard against unfavorable outcomes. Additionally, providing probabilistic forecasts as opposed to only suggesting decisions allows the operator to *incorporate domain knowledge* into the decision process.

To address the need for uncertainty estimation with nonstationary data, we consider a probabilistic forecasting task for an arbitrary (potentially adversarial) data stream. This raises the question: *How can we argue for the validity of probabilistic forecasts for events that may be unrelated to past observations, or even deterministically chosen to invalidate predictions?* One approach is to appeal to *calibration*, which says that in the long run, the observed frequency of an event should match its probability under the forecasts (Dawid, 1982; Foster &

Vohra, 1998; Vovk et al., 2005). In other words, in hindsight the data stream should resemble samples drawn from the forecasted distributions.

In this work, we study the scope of calibration guarantees that can be provided for forecasts of (potentially) adversarial data. We consider a stream of outcomes on any compact metric space, encapsulating both classification tasks and regression for bounded outcomes. Building on a stream of work connecting calibration to Blackwell approachability (Blackwell, 1956; Foster, 1999; Abernethy et al., 2011; Perchet, 2013), we model the forecasting task as a zero-sum game played between the Forecaster and Nature. We design the payoff (i.e., "win condition") of this game to measure the calibration and predictive performance of the forecasts. This allows us to give a general theoretical procedure to enforce any form of calibration that meets mild regularity conditions. This procedure and its accompanying calibration guarantees serve as a proof that most popular forms of calibration can be enforced without any assumptions about how the data stream evolves, with miscalibration decaying in time as $O(1/\sqrt{T})$. However, implementing this procedure is in some cases computationally hard. Thus, we study particular forms of calibration which can be guaranteed by tractable algorithms, and give gradient-based algorithms that offer greater generality at the cost of worse guarantees. This allows us to enforce new forms of calibration in online settings, such as decision calibration (Zhao et al., 2021b) and distribution calibration (Song et al., 2019). Furthermore, we can combine multiple notions of calibration, for example simultaneously enforcing moment-matching and quantile calibration.

Robustness to adversarial data is valuable since it alleviates the need to make unverifiable assumptions about the future. However, most real-world data is not truly adversarial: it is unlikely that nature is conspiring to choose the weather to invalidate weather forecasts. Many forecasters make assumptions that are approximately valid and, as a result, make more accurate predictions. A good forecasting strategy should be able to benefit from this domain knowledge. Thus, we also apply our methodology as a post-processing procedure to achieve no-regret recalibration for existing forecasters; given a set of expert forecasters and a scoring rule, we construct forecasts that asymptotically match the score of the best forecaster, while also guaranteeing calibration.

Our main contributions are:

1. We propose a unifying framework for calibration in online learning, describing how many forms of calibration can be expressed as payoffs in a repeated game.
2. We provide finite-sample calibration guarantees via a theoretical forecasting procedure that applies to any form of calibration fitting into our framework.
3. We introduce ORCA, a novel gradient-based algorithm to implement the forecasting procedure, and efficient algorithms optimized for specific forms of calibration.
4. We perform two empirical studies applying ORCA to recalibrate forecasters on regression tasks, demonstrating its ability to improve both calibration and downstream decisions.

## 2 Problem Setting

Consider a probabilistic forecasting task, where the goal of the Forecaster is to give a probability distribution for the next outcome in a sequence (e.g., the weather tomorrow). At each time step $t = 1, 2, \ldots$, Nature reveals features $x_t \in \mathcal{X}$ (e.g., today's temperature and air pressure). Next, Forecaster chooses a forecast $p_t$ in $\Delta(\mathcal{Y})$, the space of probability distributions over an outcome space $\mathcal{Y}$. Finally, Nature reveals the outcome $y_t \in \mathcal{Y}$. The outcome space $\mathcal{Y}$ is assumed to be a compact metric space, such as a finite set of labels in classification or a bounded interval in regression. We make no assumptions about the feature space $\mathcal{X}$.

**Online Learning** Forecaster has access to the history and the features when choosing the forecast $p_t = \text{Forecaster}(h_{t-1}, x_t)$, where $h_{t-1}$ represents the history of all data and forecasts through time $t - 1$. Nature then observes the forecast before choosing the outcome $y_t = \text{Nature}(h_{t-1}, x_t, p_t)$. We allow for the possibility that Nature acts adversarially, choosing whichever outcome is least favorable for Forecaster. In this sense, the task can be viewed as a zero-sum repeated game played between Forecaster and Nature. The generality of the online learning paradigm means our algorithms and theoretical results apply for any sequence of features and outcomes $(x_t, y_t)$ over $\mathcal{X} \times \mathcal{Y}$. In particular, our guarantees hold for nonstationary data generating processes arising due to distribution shifts, feedback loops, time series, and adversarial actors.

**Calibration**   A notion of calibration represents a property we would expect to observe if each outcome $y_t$ were sampled independently from the forecast $p_t$. For example, suppose $y_t \in \{0, 1\}$ is binary and $p_t \in [0, 1]$ is a forecasted probability that $y_t = 1$. In the batch setting, where $\{(x_t, y_t)\}_{t=1}^T$ are i.i.d. random variables (e.g., a held-out test dataset), binary distribution calibration says that $\mathbb{P}(y_t = 1 \mid p_t \approx p) \approx p$ for all forecasts $p \in \Delta(\mathcal{Y})$ that occur sufficiently often. In the online setting, outcomes are an arbitrary sequence and not necessarily random variables, so the probabilistic definition is not well-defined. Instead, we define calibration analogously via empirical frequencies

$$\lim_{T \to \infty} \frac{\sum_{t=1}^T \mathbb{1}\{y_t = 1, p_t \approx p\}}{\sum_{t=1}^T \mathbb{1}\{p_t \approx p\}} \approx p, \tag{1}$$

where the indicator variable $\mathbb{1}\{E\}$ is 1 if the event $E$ occurs and 0 otherwise (Vovk et al., 2005). The indicator $\mathbb{1}\{p_t \approx p\}$ is a continuous approximation to the function $\mathbb{1}\{|p_t - p| \leq \epsilon\}$ for small $\epsilon$ (see Section 4 for discussion). In words, Equation 1 says that when the forecast $p_t$ is close to any fixed value $p$, the fraction of times that $y_t = 1$ approaches $p$. Calibration requires that this holds for all $p \in [0, 1]$ such that $p_t \approx p$ with nonzero frequency in the limit of large $T$, i.e., $\liminf_{T \to \infty} \frac{1}{T} \sum_{t=1}^T \mathbb{1}\{p_t \approx p\} > 0$.

**Blackwell Approachability**   Blackwell's Approachability Theorem gives sufficient conditions under which a player in a repeated zero-sum game can control a vector-valued payoff, when averaged over the rounds of the game. In each round, $t = 1, 2, \dots$, two players, Forecaster and Nature, sequentially choose actions $a_t \in \mathcal{A}$ and $b_t \in \mathcal{B}$, respectively, yielding a vector-valued payoff $\pi(a_t, b_t) \in \mathbb{R}^m$. Forecaster's goal is to drive the average payoff $\overline{\pi}_T = \frac{1}{T} \sum_{t=1}^T \pi(a_t, b_t)$ towards a target set $\mathcal{E} \subseteq \mathbb{R}^m$, while Nature tries to prevent this. The distance between the average payoff and the target set is defined as $d(\overline{\pi}_T, \mathcal{E}) := \inf_{\overline{\pi} \in \mathcal{E}} \|\overline{\pi}_T - \overline{\pi}\|$. A set $\mathcal{E}$ is *approachable* if Forecaster can ensure that $d(\overline{\pi}_T, \mathcal{E}) \to 0$ as $T \to \infty$, regardless of Nature's strategy. For our purposes, the target set is the origin $\mathcal{E}_0 = \{0\}$. Blackwell (1956) showed that $\mathcal{E}_0$ is approachable if Forecaster can ensure the payoff lies in a halfspace defined by the average payoff, regardless of Nature's action:

$$\inf_{a \in \mathcal{A}} \sup_{b \in \mathcal{B}} \langle \pi(a, b), \overline{\pi} \rangle \leq 0, \quad \forall \overline{\pi} \in \mathbb{R}^m. \tag{2}$$

## 3   The Forecasting Game

In this section, we express popular forms of calibration as the average payoff in a zero-sum game played between Forecaster and Nature. Forecaster's actions are distributions $p \in \Delta(\mathcal{Y})$, and Nature's actions are outcomes $y \in \mathcal{Y}$. In subsequent sections, we show how to achieve these notions of calibration from a unified theoretical (Section 4) and algorithmic (Section 5) perspective.

---
**Procedure 1** Probabilistic Forecasting
---
    **for** $t = 1, 2, \dots$ **do**
        Nature reveals features $x_t \in \mathcal{X}$
        Forecaster predicts a distribution $p_t \in \Delta(\mathcal{Y})$
        Nature reveals an outcome $y_t \in \mathcal{Y}$
        Forecaster receives payoff $\pi(x_t, p_t, y_t) \in \mathcal{H}$
    **end for**
---

The payoff $\pi(x_t, p_t, y_t)$ reflects the error of the forecast $p_t$ for features $x_t$ and outcome $y_t$, and takes values in an arbitrary Hilbert space $\mathcal{H}$ (e.g., a scalar or a vector). The average payoff up to time $T$ is denoted as $\overline{\pi}_T = \frac{1}{T} \sum_{t=1}^T \pi(x_t, p_t, y_t)$. We are interested in the inner product norm $\|\overline{\pi}_T\| = \sqrt{\langle \overline{\pi}_T, \overline{\pi}_T \rangle_{\mathcal{H}}}$, which for vector-valued payoffs is the familiar Euclidean norm.

**Definition 3.1.** The *miscalibration* of forecasts $p_1, p_2, \dots, p_T$ for features $x_1, x_2, \dots, x_T$ and outcomes $y_1, y_2, \dots, y_T$ with respect to a payoff $\pi$ is the norm of the average payoff, $\|\overline{\pi}_T\|$. A sequence of forecasts is *calibrated* if the miscalibration vanishes, that is $\|\overline{\pi}_T\| \to 0$ as $T \to \infty$.

We now give concrete examples of how calibration can be encoded as a payoff to demonstrate the versatility of Procedure 1 as an abstraction for calibration.

**Binary Calibration**   As a simple example, consider a binary label indicating whether it rains ($y = 1$) or not ($y = 0$). Forecaster wants to ensure that on days she predicts rain with probability $p$, the observed frequency of rain is also $p$. We use a vector payoff indexed by a discrete set of probabilities $p \in [0, 1]$, with entries $\pi_p(x_t, p_t, y_t) = \mathbb{1}\{p_t \approx p\}(y_t - p)$. This induces the miscalibration measure $\|\bar{\pi}_T\|^2 = \sum_p \left(\frac{n_{p,T}}{T}\right)^2 (f_{p,T} - p)^2$, where $n_{p,T}$ is the number of days $p_t \approx p$ and $f_{p,T}$ is the frequency of the event $y_t = 1$ when $p_t \approx p$, up to day $T$. This closely resembles the Expected Calibration Error (Guo et al., 2017), except using an L2 norm on the errors instead of an L1 norm.

**Quantile Calibration**   In a regression setting, let $p_t$ be the predicted distribution for the daily high temperature $y_t \in \mathbb{R}$. Quantile calibration requires that for any quantile $q \in [0, 1]$ (e.g., the median), the proportion of days $y_t$ falls below the $q$-quantile of $p_t$ is $q$ (e.g., half the time). We use a vector payoff indexed by a discrete set of quantiles $q$, given by $\pi_q(x_t, p_t, y_t) = \mathbb{1}\{F_{p_t}(y_t) \le q\} - q$, where $F_{p_t}$ is the cdf of $p_t$. The induced miscalibration measure, known as the Quantile Calibration Error (Kuleshov et al., 2018), is $\|\bar{\pi}_T\|^2 = \sum_q (f_{q,T} - q)^2$, where $f_{q,T}$ is the frequency of the event $F_{p_t}(y_t) \le q$ up to day $T$.

**Moment Matching**   Beyond binary and quantile calibration, different notions of faithfulness are relevant depending on the intended use or interpretation of the forecast. For example, if temperature forecasts are used to predict the mean or variability of the temperature over a period, it is crucial for the averaged moments of the forecasts to match those of the data. This can be measured using the payoff $\pi_k(x_t, p_t, y_t) = \mathbb{E}_{y \sim p_t}[y^k] - y_t^k$, indexed by moments $k \in \{1, 2\}$. The induced miscalibration metric $\|\bar{\pi}_T\|^2$ measures the squared error in the forecasted moments. We can match additional moments of the forecasts and data with $k > 2$, eventually requiring that the marginal distributions match exactly when $k$ goes to $\infty$.

**Decision Calibration**   Forecasts directly influence decision-making in areas such as adaptive experimental design (Deshpande et al., 2024), model predictive control (Garcia et al., 1989), and model-based reinforcement learning (Malik et al., 2019). In these scenarios, decision-makers must trust that forecasts accurately reflect the utility of each available action $a \in \mathcal{A}$. For instance, to evaluate whether forecasts systematically undervalue or overvalue any action, we can use the payoff $\pi_a(x_t, p_t, y_t) = \mathbb{E}_{y \sim p_t} u(a, y, x_t) - u(a, y_t, x_t)$, indexed by actions $a \in \mathcal{A}$, where $u(a, y, x)$ is the utility of action $a$ with outcome $y$ in context $x$. A decision-maker should also be concerned if she experiences *swap regret* (Blum & Mansour, 2007), wherein she could increase her utility by swapping all occurrences of some action $a$ with another action $a'$. This possibility can be measured using the payoff indexed by pairs of actions $(a, a')$, given by $\pi_{a,a'}(x_t, p_t, y_t) = \mathbb{1}\{a' = \delta(p_t, x_t)\}(\mathbb{E}_{y \sim p_t}[u(a, y, x_t)] - u(a, y_t, x_t))$, where $\delta(p_t, x_t)$ is the Bayes optimal action when $y$ is distributed according to $p_t$ with context $x_t$. By controlling these notions of calibration, Forecaster can accurately evaluate the average utility of each action and make decisions with vanishing swap regret.

**A General Recipe**   The above payoffs have a shared structure, which can be abstracted into a general payoff function of the form $\pi(x, p, y) = c(x, p) \otimes (\mathbb{E}_{y' \sim p}[w(y')] - w(y))$, where $c$ and $w$ are vector-valued functions and $\otimes$ is the outer product. Here, $w$ is a "witness function" specifying the quantities that should match on average between the forecasts and outcomes, and $c$ is a "context function" specifying the conditioning events under which the quantities should match. Under weak conditions (see Appendix A), calibration can be achieved online for any payoff of this form.

## 4   Winning the Forecasting Game with Provable Calibration

In the previous section, we described how calibration can be expressed using payoff functions. In this section, we use this framework to describe a general forecasting strategy that guarantees miscalibration decreases as $O(1/\sqrt{T})$. Furthermore, we provide a unified approach to enforce multiple forms of calibration simultaneously. Our strategy requires solving an optimization problem which is computationally hard for some forms of calibration. In Section 5, we propose an inexact but tractable approach using gradient-based optimization.

Our work builds on a recent stream of work that leverages Blackwell approachability and related fixed-point theorems to study calibration in online learning (Perchet, 2013; Vovk et al., 2005; Abernethy et al., 2011; Lee et al., 2022; Gupta & Ramdas, 2022; Noarov & Roth, 2023). Our primary theoretical contributions are twofold.

In Section 4.1, we establish general conditions for achieving calibration, accommodating arbitrary compact outcome spaces and nonconvex calibration metrics. In Section 4.2, we describe a framework to simultaneously achieve multiple measures of calibration and performance with a unified optimization procedure. Proofs for all results can be found in Appendix A.

## 4.1 An Existence Result for Online Calibration

First, we introduce three mild assumptions about the payoff function that underpin our main results.

**Condition 1** (Boundedness). *The norm of the payoff is bounded by $B := \sup_{x \in \mathcal{X}, p \in \Delta(\mathcal{Y}), y \in \mathcal{Y}} \|\pi(x, p, y)\|^2 < \infty$.*

**Condition 2** (Consistency). *The expected payoff is zero when the outcome is drawn from the forecast:* $\mathbb{E}_{y \sim p}[\pi(x, p, y)] = 0, \forall x \in \mathcal{X}, \forall p \in \Delta(\mathcal{Y})$.

**Condition 3** (Continuity). *The payoff is continuous as a function of the forecast $p \mapsto \pi(x, p, y)$ on $\Delta(\mathcal{Y})$ under the Wasserstein metric, $\forall x \in \mathcal{X}, \forall y \in \mathcal{Y}$.*

The first condition ensures that a single bad prediction has bounded impact on the cumulative miscalibration. The second condition encodes the requirement that a calibration measure is optimized by a perfect forecast. The third condition requires that the payoff is continuous. For any discontinuous payoff, continuity can be achieved by smoothing the payoff on an arbitrarily small scale.

Our main result follows from two observations. The first observation is a simple argument that bounds miscalibration in terms of the inner product between the current and average payoff.

**Proposition 4.1.** *If the payoff is bounded (Condition 1), then*

$$\|\overline{\pi}_T\|^2 \le \frac{B}{T} + \frac{2}{T} \sum_{t=1}^{T} \langle \overline{\pi}_{t-1}, \pi(x_t, p_t, y_t) \rangle. \tag{3}$$

Proposition 4.1 implies that if we can ensure the inner product in Equation 3 is nonpositive, then miscalibration will decrease as $\|\overline{\pi}_T\|^2 \le \frac{B}{T}$. The second observation guarantees that this inner product can be made nonpositive, regardless of Nature's play.

**Proposition 4.2.** *If the payoff is consistent (Condition 2) and continuous (Condition 3), then for all $\overline{\pi} \in \mathcal{H}$ and $x \in \mathcal{X}$, there exists a forecast $p \in \Delta(\mathcal{Y})$ such that*

$$\max_{y \in \mathcal{Y}} \langle \overline{\pi}, \pi(x, p, y) \rangle \le 0. \tag{4}$$

Proposition 4.2 is proven by using continuity to invoke the Ky Fan minimax inequality, and consistency to ensure that the resulting bound is nonpositive. Together, these results suggest a forecasting strategy which we formalize in Algorithm 1: play forecasts satisfying the halfspace condition of Equation 4, making the second term of Equation 3 nonpositive, and guaranteeing that $\|\overline{\pi}_T\|^2 \le \frac{B}{T}$.

Identifying the specific forecast that satisfies the halfspace inequality Equation 4 can be difficult. For now, we assume access to an oracle HalfSpaceOracle($\overline{\pi}, x$) that returns a forecast $p$ satisfying Equation 4. This oracle is tasked with solving the following minimax optimization problem over $p$:

$$\min_{p \in \Delta(\mathcal{Y})} \max_{y \in \mathcal{Y}} \langle \pi(x_t, p, y), \overline{\pi}_t \rangle, \tag{5}$$

whose optimal solution $p_*$ is guaranteed by Proposition 4.2 to achieve $\langle \pi(x_t, p_*, y_t), \overline{\pi}_t \rangle \le 0$. We defer the implementation of the oracle to Section 5.

**Theorem 4.3.** *If the payoff satisfies Conditions 1-3, then Algorithm 1 has miscalibration bounded by $\|\overline{\pi}_T\|_{\mathcal{H}}^2 \le B/T$ for any sequence $\{(x_t, y_t)\}_{t=1}^{\infty}$.*

Theorem 4.3 follows directly from Propositions 4.1 and 4.2, and gives a finite-sample calibration guarantee. Importantly, we make no convexity assumptions about the payoff, and allow for payoffs in a Hilbert space, which is useful for infinite dimensional payoffs that arise in regression. Thus, this generalizes existing results that require discrete outcomes and a bilinear payoff (e.g., Vovk et al., 2005; Kakade & Foster, 2008). In conclusion, we can achieve general notions of calibration for nonstationary data caused by feedback loops, distribution shifts, time series, and adversarial environments.

---

**Algorithm 1** Blackwell Forecasting

---

**Require:** Payoff $\pi$, HalfSpaceOracle

    $\overline{\pi}_0 \leftarrow 0_{\mathcal{H}}$                                                   *// Initialize avg payoff as zero*

    **for** $t = 1, 2, \ldots$ **do**

        Nature reveals features $x_t \in \mathcal{X}$

        Query HalfSpaceOracle($\overline{\pi}_{t-1}, x_t$) for a distribution $p_t$ that satisfies the inequality $\max_{y \in \mathcal{Y}} \langle \overline{\pi}_{t-1}, \pi(x_t, p_t, y) \rangle \leq 0$

        Announce forecast $p_t$, observe outcome $y_t$, and receive payoff $\pi(x_t, p_t, y_t)$

        $\overline{\pi}_t \leftarrow \frac{t-1}{t}\overline{\pi}_t + \frac{1}{t}\pi(x_t, p_t, y_t)$                                         *// Update avg payoff*

    **end for**

---

## 4.2 Multi-Objective and No-Regret Recalibration

Robustness to adversarial data allows us to avoid making assumptions about the data generating process. However, in practice most forecasting tasks are not truly adversarial; Nature is most likely not conspiring for the weather to contradict weather forecasts. In this section, we describe how to apply Algorithm 1 as a post-processing step for another forecaster. This allows us to take advantage of the inductive biases and valid assumptions of other forecasters, while maintaining guarantees if those assumptions fail. Our work extends recalibration techniques from the offline setting (e.g., Kuleshov et al., 2018; Marx et al., 2024) to nonstationary data. A recent stream of work on "calibeating" studies recalibration in the online setting (Foster & Hart, 2023), and perhaps most relevant to our work is Lee et al. (2022). The most significant differences between Lee et al. (2022) and our work are that we give deterministic forecasts and do not require discretization.

Specifically, our goal is to guarantee calibration while simultaneously achieving loss comparable to known expert forecasters. We begin by defining a standard notion of regret. Let $\ell(p, y)$ be a bounded and continuous real-valued proper scoring rule, where smaller values indicate a better forecast. We are given $k$ expert forecasters, whose forecasts at each step are provided as features $x = (x_1, \ldots, x_k)$. Here, $x_i \in \Delta(\mathcal{Y})$ is a distribution representing the forecast made by expert $i$. The regret with respect to the best performing expert is given by

$$R_T := \frac{1}{T}\sum_{t=1}^{T}\ell(p_t, y_t) - \min_{1 \leq i \leq k}\frac{1}{T}\sum_{t=1}^{T}\ell(x_{i,t}, y_t). \tag{6}$$

We can bound this regret term in the framework of Blackwell approachability via the $k$-dimensional payoff whose $i$th entry is the excess loss relative to the $i$th expert, $\pi_i^{\text{REG}}(x, p, y) = \ell(p, y) - \ell(x_i, y)$. The norm of the average payoff then satisfies

$$\left\|\overline{\pi}_T^{\text{REG}}(x, p, y)\right\|_2^2 \geq \left\|\overline{\pi}_T^{\text{REG}}(x, p, y)\right\|_\infty^2 = R_T^2 \tag{7}$$

Thus, we can treat the regret as simply another payoff that we optimize using Algorithm 1. See Appendix A.1 for the details of this comparison. To achieve calibration with a no-regret guarantee, we note that Forecaster can control multiple payoffs simultaneously with the same $O(1/\sqrt{T})$ time dependency.

**Proposition 4.4.** *Let $\pi^{(1)}, \ldots, \pi^{(n)}$ be $n$ payoff functions taking values in Hilbert spaces $\mathcal{H}_1, \ldots, \mathcal{H}_n$, respectively. Suppose each payoff $\pi^{(i)}$ satisfies the Conditions 1-3 with bound $B_i$. Applying Algorithm 1 to the direct sum payoff $\pi^{(1)} \oplus \cdots \oplus \pi^{(n)}$ ensures $\sum_{i=1}^{n}\|\overline{\pi}_T^{(i)}\|_{\mathcal{H}_i}^2 \leq \frac{1}{T}\sum_{i=1}^{n}B_i$, and using the normalized payoff $\frac{\pi^{(1)}}{B_1} \oplus \cdots \oplus \frac{\pi^{(n)}}{B_n}$ ensures $\|\overline{\pi}_T^{(i)}\|_{\mathcal{H}_i}^2 \leq \frac{nB_i}{T}$ for $1 \leq i \leq n$.*

Thus, we can match the performance — up to an $O(1/\sqrt{T})$ gap — of any expert forecaster, while guaranteeing multiple forms of calibration.

## 5 Algorithms

In the previous section, we gave calibration guarantees for a forecasting strategy that assumed access to an oracle solution to a minimax optimization problem. In this section we describe algorithms to implement this oracle. Since this optimization problem is PPAD-hard in some cases (Hazan & Kakade, 2012), an efficient algorithm must compromise on either generality or the strength of the guarantees. We consider both approaches, first giving a general gradient-based optimization strategy without strong guarantees, and then providing tractable oracles for specialized payoffs.

### 5.1 Gradient-Based Blackwell Forecasting

Tractable half-space oracles are attractive in that they guarantee calibration. However, there are two important outstanding limitations: first, some payoff functions do not admit tractable oracles, as indicated by computational hardness results (Hazan & Kakade, 2012). Second, it is not easy to combine tractable oracles for individual payoffs into oracles for multiple payoffs; given two payoff functions $\pi_1$ and $\pi_2$ each with a tractable oracle, it is not clear how to use the oracles to construct a tractable oracle for $\pi_1 \oplus \pi_2$. To mitigate these issues, we introduce Online Regression Calibration against an Adversary (ORCA), a gradient-based approach for solving the oracle minimax problem. ORCA iteratively optimizes an upper bound on the miscalibration (i.e., the minimax inner product). If that upper bound drops below zero, we can control miscalibration. If the upper bound remains positive because we fail to identify a global optimum of the nonconvex optimization problem, then we play the best identified forecast, with miscalibration limited by the identified upper bound. Thus, while ORCA does not guarantee calibration, we know *before observing the outcome* whether it is possible for the miscalibration to worsen. Additionally, we can easily combine multiple payoffs by concatenating them, without needing to adjust the optimization algorithm.

**Parameterizing the Forecasts**   We introduce two parameterized families of distributions over the outcome space: one for the forecast distributions and one for the adversarial outcome distributions. The forecast family is given by $\mathcal{P} = \{p_\theta : \theta \in \Theta\}$, and should be flexible enough to approximate any outcome distribution to guarantee the existence of a solution to the half-space oracle problem. For example $\mathcal{P}$ could be a neural network that represents the PDF or CDF of the forecast, or a parametric family of mixture distributions (e.g., $\theta$ encodes the means, variances, and mixture weights of a Gaussian mixture). Note that the parameters $\theta$ describe a *distribution $p_\theta$* over the outcome space, not model weights mapping inputs to a distribution.

To enable efficient gradient-based optimization, we give Nature the option to play a randomized outcome, represented by a distribution $q \in \Delta(\mathcal{Y})$. Since Nature plays last in each round, this does not affect the optimal solution to the minimax problem. The adversary family $\mathcal{Q} = \{q_\phi = \sum_{k=1}^{K} \phi_k q_k : \phi \in \mathbb{R}^K, \phi \geq 0, \phi^\top \mathbf{1} = 1\}$ is chosen to be a mixture distribution with fixed components $(q_1, \ldots, q_K)$ for $q_k \in \Delta(\mathcal{Y})$, and flexible mixture weights $(\phi_1, \ldots, \phi_K)$. Similarly, the adversary family should be flexible enough to approximate any distribution over the outcome space, to ensure the oracle does not underestimate the worst-case miscalibration.

**Gradient-Based Optimization**   Each single forecast requires solving an optimization problem over the forecast family. However, note that this optimization is over a relatively small space—on the order of the 10's or 100's of parameters composing $\theta$. Using the above parameterizations, we write the half-space oracle task (Equation 5) as

$$\min_{p \in \Delta(\mathcal{Y})} \max_{y \in \mathcal{Y}} \langle \overline{\pi}_{t-1}, \pi(x_t, p, y) \rangle \quad = \min_{p \in \Delta(\mathcal{Y})} \max_{q \in \Delta(\mathcal{Y})} \mathbb{E}_{y \sim q} \left[ \langle \overline{\pi}_{t-1}, \pi(x_t, p, y) \rangle \right] \tag{8}$$

$$\approx \min_{\theta \in \Theta} \max_{\phi \in \Delta_{k-1}} \sum_{k=1}^{K} \phi_k \mathbb{E}_{y \sim q_k} \left[ \langle \overline{\pi}_{t-1}, \pi(x_t, p_\theta, y) \rangle \right] \tag{9}$$

where $\Delta_{k-1} = \{\phi \in \mathbb{R}^k : \phi \geq 0, \phi^\top 1 = 1\}$ is the probability simplex. The first equality holds because the worst-case outcome is equivalent to the worst-case outcome distribution, due to the linearity of the expectation. The second (approximate) equality replaces the full space of distributions with the parameterized families we will optimize over. When the components $q_k$ are Dirac measures (making $q_\phi$ a Categorical distribution), we can compute the expectation analytically. In general, we can also approximate the expectation using Monte Carlo simulation. Now, the inner maximization problem is a linear program (LP) in the mixture weights $\phi$,

which we can solve in $O(K^{3.5})$ time (Amos & Kolter, 2017). When solving the outer minimization problem, we differentiate through the LP solution using differentiable convex optimization solvers (Agrawal et al., 2019). We now have a minimization task over $\theta$ with a differentiable objective, which we can optimize using standard gradient-based optimizers such as Adam (Kingma & Ba, 2014). The full forecasting strategy for ORCA is summarized in Algorithm 2.

**The Impacts of Approximation** Since exact optimization of Equation 8 is intractable, in Equation 9 we replaced the sets over which we perform the maximization (Nature's task) and the minimization (Forecaster's task) with parameterized families. The approximation of the maximization weakens Nature, which may lead us to underestimate the cost of the worst-case outcome. However, the impact of this approximation can be bounded. Let Nature's optimal score be denoted by $A^* := \max_{q \in \Delta(\mathcal{Y})} \mathbb{E}_{y \sim q} [\langle \overline{\pi}_{t-1}, \pi(x_t, p, y) \rangle]$, and Nature's optimal *realizable* score be denoted by $A^*_{\mathcal{Q}} := \max_{\phi \in \Delta_{k-1}} \sum_{k=1}^{K} \phi_k \mathbb{E}_{y \sim q_k} [\langle \overline{\pi}_{t-1}, \pi(x_t, p, y) \rangle]$. Let $S_k$ be the support of each distribution $q_k$ in Equation 9. Denote the greatest diameter of these supports by $d = \max_{k=1,\ldots,K} \text{diam}(S_k)$, and denote their union by $S = \cup_{k=1}^{K} S_k$. Suppose that $S$ is an $\epsilon$-net of $\mathcal{Y}$, meaning that for all $y \in \mathcal{Y}$ there exists some $s \in S$ such that $\|y - s\| \leq \epsilon$. Additionally, suppose that the payoff $\pi(x, p, y)$ is $L$-Lipschitz in $y$ for all $x \in \mathcal{X}$ and $p \in \Delta(\mathcal{Y})$. Then, it is easy to show that $0 \leq A^* - A^*_{\mathcal{Q}} \leq (d + \epsilon) L \cdot \|\overline{\pi}_{t-1}\|$. This bound controls the error in our estimate of the worst-case miscalibration for any particular forecast.

Conversely, the approximation of the minimization in Equation 9 weakens Forecaster. If the forecast family is not sufficiently expressive, then we may overlook the best forecast. However, note that this approximation does not weaken Nature, so while this approximation can prevent us from identifying the optimal worst-case miscalibration guarantee, the validity of the guarantee we do identify is preserved.

---

**Algorithm 2** ORCA: Gradient-Based Blackwell Forecasting

---

**Require:** Payoff $\pi$, Forecast set $\mathcal{P}$, Adversary set $\mathcal{Q}$

  $\overline{\pi}_0 \leftarrow 0_{\mathcal{H}}$                                                         *// Initialize avg payoff as zero*

  **for** $t = 1, 2, \ldots$ **do**

    Nature reveals features $x_t \in \mathcal{X}$

    Initialize $p_{\theta_t} \in \mathcal{P}$

    **while** not converged **do**

      Compute $\ell_t(\theta_t; \mathcal{Q}, \pi)$, the worst-case miscalibration for $p_{\theta_t}$ from the LP solver.    *// Equation 9*

      Update $\theta_t$ using gradient-based optimization to minimize $\ell_t(\theta_t; \mathcal{Q})$

    **end while**

    Announce forecast $p_{\theta_t}$, observe outcome $y_t$, and receive payoff $\pi(x_t, p_t, y_t)$

    $\overline{\pi}_t \leftarrow \frac{t-1}{t} \overline{\pi}_t + \frac{1}{t} \pi(x_t, p_{\theta_t}, y_t)$                           *// Update avg payoff*

  **end for**

---

## 5.2 Algorithms for Specialized Notions of Calibration

Algorithm 2 is easy to implement, works with any differentiable payoff, supports no-regret recalibration, and performs well in practice. For many definitions of calibration, we can also design specific half-space oracles that provably achieve the desired notion of calibration, but they need to be handcrafted for each calibration type (see Appendix C).

**Theorem 5.1.** *There exist half-space oracles for quantile calibration, distribution calibration, moment-based calibration, and decision calibration, which provably solve the half-space problem.*

The time and space complexity for each algorithm is typically $O(1/\epsilon^k)$ for some $k > 0$ (e.g., the number of moments); see Appendix C. In simple settings (binary calibration and adaptive conformal inference), we provide $O(\log(1/\epsilon^k))$ algorithms. In general, the exponential dependence on $k$ is inevitable, as even for $k$-class classification, the optimization problem is PPAD-hard (Hazan & Kakade, 2012).

Additionally, in Appendix D we include results for specialized algorithms for no-regret recalibration, which guarantee convergence at the cost of added complexity.

|  | wind | | sunspot | |
| Forecaster | QCE | SMAPE | QCE | SMAPE |
| --- | --- | --- | --- | --- |
| stochastic gradient trees | 0.163 | **0.403** | 0.100 | **0.505** |
| + isotonic | 0.246 | 0.554 | 0.242 | 1.081 |
| + ORCA (ours) | **0.013** | 0.405 | **0.045** | 0.513 |
| neural network | 0.142 | **0.353** | 0.048 | **0.447** |
| + isotonic | 0.121 | 0.567 | 0.147 | 1.061 |
| + ORCA (ours) | **0.083** | 0.397 | **0.042** | 0.453 |
| Hoeffding tree | 0.212 | 0.545 | 0.072 | **0.432** |
| + isotonic | 0.186 | **0.428** | 0.116 | 1.061 |
| + ORCA (ours) | **0.016** | 0.431 | **0.040** | 0.479 |
| marginal | 0.126 | 0.545 | 0.046 | **0.534** |
| + isotonic | 0.094 | **0.359** | 0.084 | 1.016 |
| + ORCA (ours) | **0.017** | 0.391 | **0.040** | 0.576 |

Table 1: Comparison of two recalibration techniques, isotonic (Kuleshov et al., 2018) and ORCA (ours) on real-world data. The goal is to reduce miscalibration (QCE), while maintaining predictive performance (SMAPE). The best value is bolded and values within 10% of best are underlined.

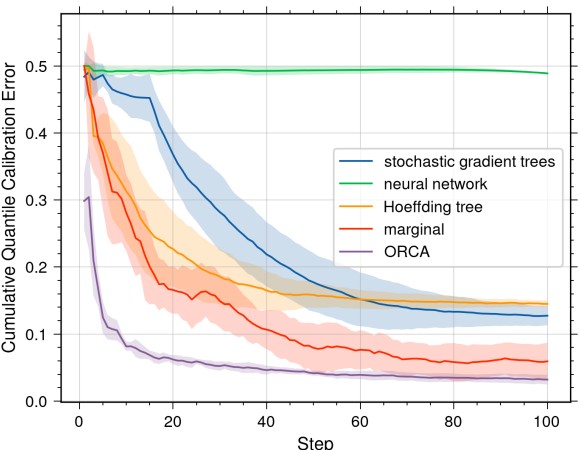

Figure 1: Comparison of cumulative quantile calibration error (QCE) for ORCA and baselines under adversarially generated data crafted to maximize QCE for each method. Results are averaged over 10 runs, with solid lines showing the mean QCE and shaded regions indicating one standard deviation.

## 6 Experiments

We perform three experiments to evaluate the ability of ORCA to recalibrate forecasts on regression tasks. In the first experiment, we test whether ORCA can improve calibration without worsening predictive performance for four different online learning models on two regression tasks. In the second experiment, we apply ORCA to data that is adversarially selected to maximize miscalibration. In the third experiment, we simulate a decision task for a wind farm operator to test whether ORCA improves downstream decision-making.

### 6.1 Recalibrating Online Learners

**Expert Forecasters** We recalibrate forecasts from four online learning algorithms implemented in the River online learning library (Montiel et al., 2021), namely stochastic gradient trees (Gouk et al., 2019), a neural network, Hoeffding adaptive trees (Bifet & Gavalda, 2009), and a naive forecaster that predicts the distribution of historical outcomes.

**ORCA Implementation** We apply ORCA to recalibrate the predictions of each expert forecaster. We parameterize the action spaces of Forecaster and Nature for the minimax optimization task both as piecewise constant densities over 50 evenly-spaced subsets of the outcome space. We find that using similar parametrizations for Forecaster and Nature is important for preventing them from leveraging limitations in each other's expressive capacity. For the payoff function, we use a combined payoff enforcing quantile calibration, no-regret for the Continuous Ranked Probability Score (CRPS) and Mean Squared Error (MSE) metrics, and moment-matching for the first two moments. We perform 400 update steps with the Adam optimizer (Kingma & Ba, 2014) to solve the minimax optimization task.

**Datasets** We consider two regression tasks. The `wind` dataset consists of hourly wind energy generation in ERCOT for the year of 2022 (of Texas, 2022). The `sunspot` dataset (Clette et al., 2014) is a standard forecasting benchmark where the task is to predict the total monthly sunspots. For each dataset, we forecast for 1000 time steps, using lag features from the previous 24 steps.

**Metrics** We evaluate predictive performance using the Symmetric Mean Absolute Percentage Error (SMAPE), defined as $\text{SMAPE} = \frac{1}{T} \sum_{t=1}^{T} \frac{|y_t - \hat{y}_t|}{(|y_t| + |\hat{y}_t|)/2}$ where $\hat{y}_t$ is the mean of the forecast $p_t$. We evaluate calibration using the Quantile Calibration Error (QCE). For the set of test quantiles $Q = \{0.01, 0.02, \ldots, 0.99\}$,

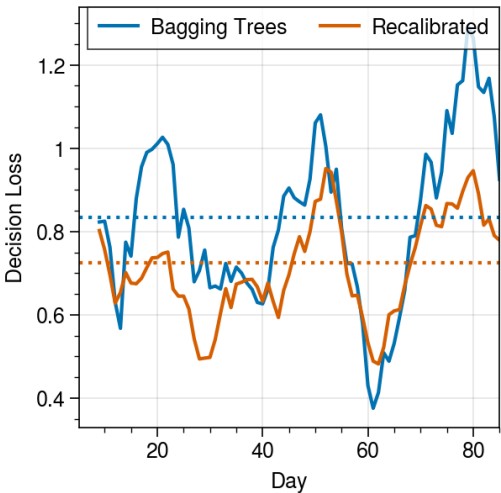

Figure 2: Comparison of the decision loss incurred by decisions based on expert forecasts before and after recalibration with ORCA. The solid line indicates the instantaneous decision loss at each step, and the dashed line indicates the mean value over the entire period. Lower values are better.

the QCE is defined as $\text{QCE} = \sum_{q \in Q} (f_{q,T} - q)^2$, where $f_{q,T}$ is the frequency of the event $F_{p_t}(y_t) \leq q$ up to day $T$. The QCE checks whether $y_t$ exceeds each quantile of the forecasts with the expected frequency.

**Baseline Comparison** We compare ORCA to isotonic recalibration (Kuleshov et al., 2018), a popular technique for recalibrating predictions in a batch setting. At each time step $t$, we fit an isotonic recalibrator based on the data up to time $t$ and apply it to recalibrate the expert forecast at time $t + 1$.

**Results** The experimental results are displayed in Table 1. We find that ORCA significantly improved calibration error, while minimally impacting predictive performance. Applying ORCA improved calibration in all 8 of the settings we tested, decreasing QCE by between 13% and 92% depending on the dataset and model. ORCA minimally impacted SMAPE, increasing SMAPE by less than 10% in 7 out of 8 cases, and even decreasing it in 2 cases. ORCA consistently outperformed isotonic recalibration in terms of QCE across all datasets and models.

## 6.2 Adversarial Prediction Task

In addition to real-world datasets, we evaluate ORCA on adversarially generated data. We consider a regression task for outcomes in the unit interval $\mathcal{Y} = [0, 1]$, and with no features $x_t$. In each round $t$, Forecaster predicts a distribution $p_t \in \Delta(\mathcal{Y})$, then Nature deterministically chooses the outcome $y_t$ to maximize the quantile calibration error (QCE) for that forecast. The goal of the forecaster is to minimize the QCE, so this is a zero-sum game between Forecaster and Nature.

Whereas ORCA is used to recalibrate other forecasters in Section 6.1, here ORCA receives no other forecast as input. The forecast parameterization is the same as in Section 6.1 and the payoff function enforces quantile calibration error. The results of the adversarial prediction experiment are shown in Figure 1. We observe that ORCA consistently achieves lower QCE than the baselines we consider.

## 6.3 Wind Farm Decision Task

In this experiment, we demonstrate how ORCA can be used to inform downstream decisions with decision calibration. We design an experiment similar to (Zhao et al., 2021a), in which a wind farm operator submits a future generation plan to a grid operator each day. On each day, the operator announces a commitment $a_t^{(i)} \in \mathbb{R}$ for each hour $i = 1, \ldots, 24$ of the next 24-hour period, and receives a penalty depending on the deviation between the offered and actual generation, given by $\ell(a_t, y_t) = \sum_{i=1}^{24} (1+\lambda)(y_t^{(i)} - a_t^{(i)})^+ + (1-\lambda)(a_t^{(i)} - y_t^{(i)})^+$.

Given a probabilistic forecast for energy generation over each of the next 24 hours, we choose the set of commitments that minimize the expected loss under the forecasted distributions. We then compare the decision losses incurred by a bagging trees forecaster to those incurred by forecasts recalibrated using ORCA. The results of the decision-making experiment are displayed in Figure 2. We observe that applying ORCA consistently improves the downstream decision loss.

## 7 Related Work

Calibration has been studied in both online and offline machine learning. In the offline setting, calibration measures have been designed for a variety of data types and applications, including quantile calibration (Kuleshov et al., 2018), binary calibration (Song et al., 2019), decision calibration (Zhao et al., 2021b), threshold calibration (Sahoo et al., 2021), marginal calibration (Gneiting et al., 2007), kernel calibration (Widmann et al., 2022), and local calibration (Luo et al., 2022). Conversely, existing work in the online setting focuses on binary calibration, with a few notable exceptions discussed below. Our work can be viewed as extending the breadth of calibration measures defined in the offline setting to the online setting.

Interest in calibrated online forecasting dates back to the 1980's (Dawid, 1982). An existing line of work studies calibration for classification tasks (Abernethy et al., 2011; Gupta & Ramdas, 2022; Okoroafor et al., 2023; Foster & Vohra, 1998; Kakade & Foster, 2008; Perchet, 2013; Vovk et al., 2005; Kuleshov & Ermon, 2017), where connections between calibration and Blackwell's approachability theorem (Blackwell, 1956) are well-known. Furthermore, Okoroafor et al. (2023) give methods for no-regret recalibration in classification. Foster & Hart (2023) introduce *calibeating*, a method for improving calibration while maintaining sharpness, therefore improving performance as measured by a proper scoring rule. Many previous analyses (e.g., Okoroafor et al., 2023; Abernethy et al., 2011; Gupta & Ramdas, 2022; Perchet, 2013) leverage the bilinearity of the payoff function in classification tasks to invoke a fixed point theorem such as von Neumann's minimax theorem (Von Neumann & Morgenstern, 1947) or Sion's minimax theorem (Sion, 1958). Similarly, the proposed algorithms in these works tend to be specialized to a specific form of calibration and data. In contrast, we study nonconvex payoffs, such as for quantile calibration in regression, necessitating the new existence criteria we derive. Furthermore, computational hardness results in this general setting motivate ORCA, our gradient-based calibration method. ORCA can easily be applied to any differentiable payoff or combination of payoffs, making it more broadly applicable than existing specialized methods.

Noarov & Roth (2023) is also closely related to our work, as they study a general class of calibration measures that satisfy an elicitability criterion. Two important distinctions between our work and Noarov & Roth (2023) are that they require randomized forecasts while we give deterministic forecasts, and they discretize the forecast space while we use a continuous forecast space. Other recent work extends online calibration to include multicalibration (Gupta et al., 2021; Bastani et al., 2022; Lee et al., 2022; Garg et al., 2024) and conformal prediction (Gibbs & Candes, 2021; Angelopoulos et al., 2023).

## 8 Conclusion

This work presents a novel framework for calibrated probabilistic forecasting for sequences over any compact outcome space. To implement our framework, we introduce generally-applicable gradient-based algorithms and provide specialized algorithms for common forecasting scenarios. Empirically, we find that our methods improve calibration and decision-making when used for recalibration in energy systems. Investigating the connections between Blackwell forecasting, multicalibration (Gupta et al., 2021; Noarov & Roth, 2023) and omnicalibration (Garg et al., 2024) is an interesting direction for future work.

**Limitations** Some common calibration metrics do not satisfy Conditions 1-3. For example, the Expected Calibration Error (Guo et al., 2017) is discontinuous due to the binning of the forecasts, and no-regret recalibration with respect to the negative log likelihood yields an unbounded payoff. Discontinuity issues can be mitigated by applying miniscule kernel smoothing to the payoff function (e.g., assigning a forecast near a discontinuity as partially in each neighboring bin) and boundedness issues can be mitigated by clipping.

Additionally, while our theoretical results apply to nondifferentiable payoffs, in practice ORCA requires differentiability to facilitate gradient-based optimization. Smoothing nondifferentiable payoffs creates a deviation between the objective being optimized and the true calibration objective. In practice, we find that this distinction is small enough that performance is not significantly impacted.

**Broader Impacts** Uncertainty estimates, and particularly those that hold under weak assumptions, support the responsible and safe deployment of machine learning systems. We hope this work enables broader adoption of uncertainty-aware decision processes by providing generally applicable calibration methods. Still, calibration is a property of collections of predictions, and should be used cautiously when considering an individual prediction.

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

# A   Proofs

We first prove two useful lemmas.

**Proposition A.1.** *If the payoff is bounded (Condition 1), then*

$$\left\|\overline{\pi}_T\right\|^2 \le \frac{B}{T} + \frac{2}{T}\sum_{t=1}^{T}\left\langle\overline{\pi}_{t-1}, \pi(x_t, p_t, y_t)\right\rangle. \tag{3}$$

*Proof.* We proceed by deriving a recursive form for $\left\|\overline{\pi}_T\right\|_{\mathcal{H}}^2$.

$$\left\|\overline{\pi}_T\right\|_{\mathcal{H}}^2 = \left\|\frac{T-1}{T}\overline{\pi}_{T-1} + \frac{1}{T}\pi_T\right\|_{\mathcal{H}}^2 \tag{10}$$

$$= \left(\frac{T-1}{T}\right)^2\left\|\overline{\pi}_{T-1}\right\|_{\mathcal{H}}^2 + \frac{1}{T^2}\left\|\pi_T\right\|_{\mathcal{H}}^2 + \frac{2(T-1)}{T^2}\left\langle\overline{\pi}_{T-1}, \pi_T\right\rangle_{\mathcal{H}} \tag{11}$$

Multiplying through by $T^2$, we have

$$T^2\left\|\overline{\pi}_T\right\|_{\mathcal{H}}^2 = (T-1)^2\left\|\overline{\pi}_{T-1}\right\|_{\mathcal{H}}^2 + \left\|\pi_T\right\|_{\mathcal{H}}^2 + 2(T-1)\left\langle\overline{\pi}_{T-1}, \pi_T\right\rangle_{\mathcal{H}} \tag{12}$$

Note that this is a recursion over $\rho_T := T^2\left\|\overline{\pi}_T\right\|_{\mathcal{H}}^2$. Unrolling this recursion gives

$$T^2\left\|\overline{\pi}_T\right\|_{\mathcal{H}}^2 = \sum_{t=1}^{T}\left\|\pi_t\right\|_{\mathcal{H}}^2 + 2\sum_{t=1}^{T}(t-1)\left\langle\overline{\pi}_{t-1}, \pi_t\right\rangle_{\mathcal{H}} \tag{13}$$

Finally, dividing through by $T^2$ and using the boundedness of $\pi_T$ gives the desired result:

$$\left\|\overline{\pi}_T\right\|_{\mathcal{H}}^2 = \frac{1}{T^2}\sum_{t=1}^{T}\left\|\pi_t\right\|_{\mathcal{H}}^2 + \frac{2}{T^2}\sum_{t=1}^{T}(t-1)\left\langle\overline{\pi}_{t-1}, \pi_t\right\rangle_{\mathcal{H}} \tag{14}$$

$$\le \frac{B}{T} + \frac{2}{T}\sum_{t=1}^{T}\left\langle\overline{\pi}_{t-1}, \pi_t\right\rangle_{\mathcal{H}} \tag{15}$$

$\square$

**Proposition A.2.** *If the payoff is consistent (Condition 2) and continuous (Condition 3), then for all $\overline{\pi} \in \mathcal{H}$ and $x \in \mathcal{X}$, there exists a forecast $p \in \Delta(\mathcal{Y})$ such that*

$$\max_{y \in \mathcal{Y}}\left\langle\overline{\pi}, \pi(x, p, y)\right\rangle \le 0. \tag{4}$$

*Proof.* Abusing notation, we write the expected payoff for an outcome $y$ distributed according to a distribution $q \in \Delta(\mathcal{Y})$ as

$$\pi(x, p, q) := \mathbb{E}_{y \sim q}\left[\pi\left(x, p, y\right)\right]. \tag{16}$$

The only randomness in Equation equation 16 is due to $y$. Observe that exchanging the worst-case $y \in \mathcal{Y}$ for the worst-case distribution $q \in \Delta(\mathcal{Y})$ does not change the value of the minimax game:

$$\min_{p \in \Delta(\mathcal{Y})}\max_{y \in \mathcal{Y}}\left\langle\overline{\pi}, \pi(x, p, y)\right\rangle = \min_{p \in \Delta(\mathcal{Y})}\max_{q \in \Delta(\mathcal{Y})}\left\langle\overline{\pi}, \pi(x, p, q)\right\rangle \tag{17}$$

To see that the LHS is not greater than the RHS, note that $\pi(x, p, y)$ can be represented as $\pi(x, p, \delta_y)$, where $\delta_y$ is a Dirac distribution at $y$. To see that the RHS is not greater than the LHS, note that the objective value $\left\langle\overline{\pi}, \pi(x, p, q)\right\rangle$ is linear in $q$, so the objective value will always be maximized by a Dirac distribution.

Next, we apply the Ky Fan Minimax Inequality Fan (1972) to upper bound the RHS.

**Theorem A.3** (Ky Fan Minimax Inequality Fan (1972))**.** *Let $\Delta(\mathcal{Y})$ be a nonempty compact convex subset of a Hausdorff topological vector space, and let $g : \Delta(\mathcal{Y}) \times \Delta(\mathcal{Y}) \to \mathbb{R}$ be a function such that:*

- $p \mapsto g(p, q)$ *is lower-semicontinuous for all* $q \in \Delta(\mathcal{Y})$

- $q \mapsto g(p, q)$ *is quasi-concave for all* $p \in \Delta(\mathcal{Y})$

*Then*

$$\min_{p \in \Delta(\mathcal{Y})} \max_{q \in \Delta(\mathcal{Y})} g(p, q) \leq \max_{q \in \Delta(\mathcal{Y})} g(q, q) \tag{18}$$

We apply the Ky Fan Minimax Inequality to the function $g(p, q) := \langle \overline{\pi}, \pi(x, p, q) \rangle_{\mathcal{H}}$. Our assumption that the payoff is mean-zero implies that $\pi(x, q, q) = 0$, so for all $q \in \Delta(\mathcal{Y})$ we have $g(q, q) = \langle \overline{\pi}, \pi(x, q, q) \rangle_{\mathcal{H}} = \langle \overline{\pi}, 0 \rangle_{\mathcal{H}} = 0$.

Thus, the lemma is proven once we meet the conditions of the Ky Fan Minimax Inequality. Namely, we must show that $\Delta(\mathcal{Y})$ is a nonempty compact convex subset of a Hausdorff topological vector space, and that $g(p, q)$ is lower-semicontinuous in its first argument and quasi-concave in its second argument.

We first show that $\Delta(\mathcal{Y})$ is a nonempty compact convex subset of a Hausdorff topological vector space. The space of signed measures on $\mathcal{Y}$ is a Hausdorff topological vector space when endowed with the weak* topology. The set of probability measures $\Delta(\mathcal{Y})$ is a subset of the space of signed measures, and it is an elementary result of probability theory that this set is nonempty and convex (see, e.g., Billingsley, 2017). The compactness of $\Delta(\mathcal{Y})$ follows from our assumption that $\mathcal{Y}$ is a compact metric space (Walkden, 2016, Theorem 10.2).

All that remains is to show that $g(p, q)$ is lower-semicontinuous in its first argument and quasi-concave in its second argument. Note that $g(p, q)$ is linear, and therefore quasi-concave, in its second argument due to its construction as an expectation. Lastly, note that Wasserstein convergence is equivalent to weak* convergence on a Polish space (Villani, 2008, Theorem 6.9), so our assumption that $g(p, q)$ is Wasserstein continuous in its first argument implies lower-semicontinuity in the weak* topology. Thus, the result follows by application of the Ky Fan Minimax Inequality. $\qquad \square$

**Theorem 4.3.** *If the payoff satisfies Conditions 1-3, then Algorithm 1 has miscalibration bounded by $\|\overline{\pi}_T\|_{\mathcal{H}}^2 \leq B/T$ for any sequence $\{(x_t, y_t)\}_{t=1}^{\infty}$.*

*Proof.* The theorem follows directly from Lemmas 4.1 and 4.2. For each time step $t$, Lemma 4.2 guarantees the existence of a forecast $p_t$ such that $\max_{y \in \mathcal{Y}} \langle \overline{\pi}_t, \pi(x_t, p_t, q_t) \rangle \leq 0$. Next, Lemma 4.1 implies that any forecasting strategy which plays such a forecast $p_t$ achieves

$$\|\overline{\pi}_T\|_{\mathcal{H}}^2 \leq \frac{B}{T} + \frac{2}{T} \sum_{t=1}^{T} \langle \overline{\pi}_{t-1}, \pi_t \rangle_{\mathcal{H}}$$
$$\leq \frac{B}{T}$$

Taking the square root of both sides yields the desired result. $\qquad \square$

**Proposition A.4.** *Let $\pi^{(1)}, \ldots, \pi^{(n)}$ be $n$ payoff functions taking values in Hilbert spaces $\mathcal{H}_1, \ldots, \mathcal{H}_n$, respectively. Suppose each payoff $\pi^{(i)}$ satisfies the Conditions 1-3 with bound $B_i$. Applying Algorithm 1 to the direct sum payoff $\pi^{(1)} \oplus \cdots \oplus \pi^{(n)}$ ensures $\sum_{i=1}^{n} \|\overline{\pi}_T^{(i)}\|_{\mathcal{H}_i}^2 \leq \frac{1}{T} \sum_{i=1}^{n} B_i$, and using the normalized payoff $\frac{\pi^{(1)}}{B_1} \oplus \cdots \oplus \frac{\pi^{(n)}}{B_n}$ ensures $\|\overline{\pi}_T^{(i)}\|_{\mathcal{H}_i}^2 \leq \frac{nB_i}{T}$ for $1 \leq i \leq n$.*

*Proof.* The proposition includes two inequalities. Define the concatenated payoff $\pi^{(1:n)} := [\pi^{(1)}, \ldots, \pi^{(n)}]$ taking values in the direct sum Hilbert space $\mathcal{H}_1^n = \bigoplus_{i=1}^{n} \mathcal{H}_i$. Note that $\pi^{(1:n)}$ inherits all the conditions of Theorem 4.3 from the component payoffs, but with the payoff bound $B_1^n = \sum_{i=1}^{n} B_i$. Thus, the first inequality is a direct result of Theorem 4.3. To see the second inequality, we first define a normalized payoff $\rho^{(1:n)} := [\pi^{(1)}/B_1, \ldots, \pi^{(n)}/B_n]$ where each component payoff has norm at most 1. Applying the first

inequality, we get that $\|\bar{\rho}_T^{(1:n)}\|_{\mathcal{H}_1^n}^2 = \sum_{i=1}^n \|\bar{\pi}_T^{(i)}\|_{\mathcal{H}_i}^2/B_i \leq \frac{n}{T}$. Since each term of the sum is nonnegative, we have for all $1 \leq i \leq n$ that $\|\bar{\pi}_T^{(i)}\|_{\mathcal{H}_i}^2 \leq \frac{nB_i}{T}$. $\qquad\square$

## A.1 An Extension to the Existence Theorem for Semi-Consistent Payoffs

Note that $\pi^{\text{REG}}$ in Section 4.2 does not satisfy consistency (Condition 2), which requires that $\mathbb{E}_{y\sim p}[\pi(x,p,y)] = 0, \forall x \in \mathcal{X}, \forall p \in \Delta(\mathcal{Y})$. Thus, we introduce a weaker condition (Condition 4), which allows us to control the positive component of the excess loss, since negative excess loss (i.e., outperforming the experts) is not problematic. Note that Condition 4 applies to $\pi^{\text{REG}}$ due to the propriety of the scoring rule; the true distribution has expected loss no greater than any other forecast.

**Condition 4** (Semi-consistency). *A vector-valued payoff $\pi(x,p,y) \in \mathbb{R}^m$ is semi-consistent if $\mathbb{E}_{y\sim p}[\pi_i(x,p,y)] \leq 0$, for all $i \in \{1,\ldots,m\}$, $p \in \Delta(\mathcal{Y})$, and $x \in \mathcal{X}$.*

**Proposition A.5.** *If a vector-valued payoff function satisfies Conditions 1, 3 and 4, then Algorithm 1 achieves $\|(\bar{\pi}_T)^+\|^2 \leq B/T$ for any sequence $\{(x_t, y_t)\}_{t=1}^\infty$, where $(\bar{\pi}_T)^+ = ((\bar{\pi}_{1,T})^+,\ldots,(\bar{\pi}_{m,T})^+)$ for $(\bar{\pi}_{i,T})^+ = \max(0,\bar{\pi}_{i,T})$ is the positive part.*

The proof of Proposition A.5 is almost identical to the proof of Theorem 4.3, with the average payoff replaced by its positive part. Applying Proposition A.5 to $\pi^{\text{REG}}$ gives the no-regret guarantee that $R_T \leq \frac{\sqrt{n}B_\ell}{\sqrt{T}}$, where $B_\ell = \sup_{p,y}\ell(p,y)$. Thus, we achieve the same $O(1/\sqrt{T})$ regret rate.

## B Calibration Payoffs

Below, we introduce an additional payoff for full distribution calibration, and add details to for additional precision for the quantile calibration payoff.

**Distribution Calibration** Distribution calibration states that across all time steps $t$ with the same forecast $p$, the distribution of outcomes should match the forecast (Song et al., 2019). In the batch setting, where the outcome $y$ and forecast $p$ are random variables, distribution calibration can be written as $(y \mid p) \sim p$. For example, when forecasting the daily high temperature, on days the forecast is a Gaussian distribution with mean $20°\text{C}$ and variance $3°\text{C}$, the observed high temperatures should be roughly Gaussian with the same mean and variance. To define distribution calibration in the online setting using payoffs, we use a vector-based payoff indexed by a discrete set of distributions $p \in \Delta(\mathcal{Y})$, given by $\pi_p(x_t, p_t, y_t) = \mathbb{1}\{p_t \approx p\}(F_{p_t} - F_{\delta(y_t)})$, where $F_{p_t}$ and $F_{\delta(y_t)}$ are the cdfs for the forecast and a Dirac measure at $y_t$, respectively. Note that a fine discretization of the space of distributions $\Delta(\mathcal{Y})$ will grow exponentially in size as $\mathcal{Y}$ grows, highlighting the computational hardness of distribution calibration.

**Quantile Calibration** Since distribution calibration is often too difficult to impose in regression—primarily due to the need for a conditioning event for each distribution—quantile calibration is a popular weaker alternative, requiring that the quantiles of forecasts are accurate, on average. That is, the outcome $y_t$ should be less than or equal to the $\alpha$-quantile of the forecast with frequency $\alpha$, for all $\alpha \in [0,1]$ (e.g., $y_t$ should exceed the forecasted median half the time). Formally, quantile calibration states that $\lim_{T\to\infty} \frac{1}{T}\sum_{t=1}^T \mathbb{1}\{y_t \leq \text{Quantile}(p_t, \alpha)\} = \alpha$ for all $\alpha \in [0,1]$. Quantile calibration enforces the law of probability that the probability integral transform (PIT) of a continuous random variable follows a uniform distribution on the unit interval. In a batch setting where the outcomes are i.i.d. random variables and if the forecasts and ground-truth distributions are continuous, the payoff $\pi_\alpha(x,p,y) = \mathbb{1}\{y_t \leq \text{Quantile}(p_t,\alpha)\} - \alpha$, indexed by $\alpha \in [0,1]$, suffices to measure quantile miscalibration. To handle the case where there are discontinuities in the cdf of the forecast, we allow forecasts over a strict superset of the outcome space and linearize across discontinuities in the forecasted CDF (see Ziegel & Gneiting (2014) for a similar approach in the batch setting), yielding the payoff

$$\pi_\alpha(x,p,y) = \lambda\mathbb{1}\{F_p(y) \leq \alpha^-\} + (1-\lambda)\mathbb{1}\{F_p(y) \leq \alpha^+\} - \alpha \tag{19}$$

where $\alpha^+ = \inf\{\alpha' \in \text{Range}(F_p) : \alpha' \geq \alpha\}$, $\alpha^- = \sup\{\alpha' \in \text{Range}(F_p) \cup \{0\} : \alpha' \leq \alpha\}$, and $\lambda = (\alpha^+ - \alpha)/(\alpha^+ - \alpha^-)$

# C  Oracles

This section introduces half-space oracles designed for specific notions of calibration. Specifically, we focus on the following: adaptive conformal inference, quantile calibration, distribution calibration, moment-based distribution calibration, and decision calibration. In each setting, we introduce the definition of calibration, provide a half-space oracle, prove its correctness, and discuss its computational complexity.

## C.1  Quantile-Based Notions of Calibration for Regression Problems

We start with perhaps the simplest setting, which considers forms of calibration defined for regression problems and in which calibration is defined using quantiles. Specifically we look at two approaches of this form: adaptive conformal inference—a setting that matches exactly the definition introduced by Gibbs & Candes (2021)—as well as quantile calibration, an extension to full quantile functions.

**Preliminaries**  We introduce both quasi-randomized and deterministic algorithms. We use the term quasi-randomized to say that the randomization will be between two forecasts (our outputs) $f_1, f_2$ that satisfy $||f_1 - f_2|| < \epsilon$, for some small $\epsilon$ and according to some norm $|| \cdot ||$. Our deterministic algorithms will assume the existence of a *root finding oracle* for a function $g : \mathcal{X} \to \mathbb{R}$ on some closed and compact set $\mathcal{X}$ (e.g., the interval [0,1]) that is guaranteed to have a point $x$ such that $g(x) = 0$.

### C.1.1  Adaptive Conformal Inference

**Setup**  In the setting of adaptive conformal inference, we are given a target quantile $\beta$ as well as a quantile function $Q_t(a)$ (that could be different at each time step) that targets a continuous outcome $y_t \in \mathbb{R}$. Our goal is to choose at each time step an $\alpha_t \in [0, 1]$ such that the $Q_t(\alpha_t)$ are on average the $\beta$-th quantile within an error tolerance of $\epsilon > 0$. In other words, we want

$$\lim_{T \to \infty} \left( \frac{1}{T} \sum_{t=1}^{T} \mathbb{1} \{y_t \leq Q(\alpha_t)\} - \beta \right) \leq \epsilon.$$

**Randomized Algorithm**  We define our algorithm as follows. First, we define a finite of set of possible outputs for $\alpha_t$ over which we will perform randomization. Specifically, we define our **action set** as $A = \{a_0, a_1, \ldots, a_M\}$ where $A$ is a discretization of $[0, 1]$, i.e., $0 = \alpha_0 \leq \alpha_i < \alpha_{i+1} \leq \alpha_M = 1$. We will play distributions $p \in \Delta(A)$ over the action set. We define our **payoffs** as follows. Let $e_t(a) = \mathbb{1} \{y_t \leq Q_t(a)\}$. Let the payoff be indexed by actions $a \in A$, given by $\pi_a(p_t, y_t) = p_t(a) (e_t(a) - \beta)$, where $p_t(a)$ is the probability $p_t$ assigns to action $a$. We want the average set of payoffs to approach a ball of size $\epsilon$ centered at the origin. Note that this yields a valid quantile estimate as defined above.

In order to achieve this goal using approachability, we need to devise a half-space oracle that will output at each time step a distribution $p_t$ over $A$ such that for any vector of historical payoffs $\overline{\pi}_t$, we have $\langle \overline{\pi}_t, \pi(p_t, y) \rangle \leq 0$ for all $y$.

We construct the following **half-space oracle** for this goal, given the average payoff $\overline{\pi}_t$:

- If for all $a, \overline{\pi}_{t,a} \geq 0$, output $\alpha_0$ with probability one.

- If for all $a, \overline{\pi}_{t,a} \leq 0$, output $\alpha_M$ with probability one.

- Otherwise, there will exist an $i \in \{1, \ldots, M\}$ such that $\text{sign}(\overline{\pi}_{t,a_i}) \neq \text{sign}(\overline{\pi}_{t,a_{i+1}})$. Then choose the above $a_i, a_{i+1}$ and play with probability $p_i = |\overline{\pi}_{t,a_i}^{-1}|/(|\overline{\pi}_{t,a_i}^{-1}| + |\overline{\pi}_{t,a_{i+1}}|^{-1})$ the value $a_i$, and otherwise $a_{i+1}$. Then for all $y$, we have:

**Lemma C.1.** *Let $\epsilon > 1/M$. Then the above algorithm is a half-space oracle for a ball of radius $\epsilon$. Furthermore, the algorithm runs in $\log(\epsilon)$ time and $O(1/\epsilon)$ space.*

*Proof.* First, we explain the time and space complexity of the algorithm. First, the space complexity is $O(M) = O(\epsilon)$ because $\epsilon > 1/M$. The time complexity is $O(\log(M)) = O(\log(\epsilon))$ because we can find an $i$ such that $\text{sign}(\overline{\pi}_{t,a_i}) \neq \text{sign}(\overline{\pi}_{t,a_{i+1}})$ (or determine that none exists) in $O(\log(M))$ time using binary search.

Next, we prove that in each of the three cases above, the property of the half-space oracle is satisfied. Note that:

$$\langle \overline{\pi}_t, \pi(p_t, y) \rangle = \overline{\pi}_{t,a_i} p_t(a_i) \left( e_t(a_i) - \beta \right) + \overline{\pi}_{t,a_{i+1}} p_t(a_{i+1}) \left( e_t(a_{i+1}) - \beta \right)$$
$$= k \cdot (e_t(a_i) - e_t(a_{i+1})) \text{ for a constant } k > 0, \text{ by choice of } p_t$$

note again that $e_t(a_1) = 1 \implies e_t(a_2) = 1$ if $a_1 < a_2$ therefore the above expression is $\langle \overline{\pi}_t, \pi(p_t, y) \rangle \leq 0$. $\square$

**Deterministic Algorithm** Our deterministic algorithm is a straight-forward extension of the above procedure. We now let the action space $A = [0, 1]$ be equal to the entire unit interval. The payoffs are the same as above for each $a \in A$. The algorithm is defined as follows. Denote the average payoff by $c : A \to \mathbb{R}$. Then we run the zero-finding oracle and output the zero of $c$ if it exists. Otherwise, either $c \geq 0$, and we output 0 or $c \leq 0$, in which case we output 1.

**Lemma C.2.** *The above algorithm is a deterministic half-space oracle for a ball of radius $\epsilon$. Its runtime is equal to that of the root-finding oracle.*

*Proof.* First, note that the above three cases in the algorithm are exhaustive for the same reason as in the previous theorem. The two edge cases are handled as before. In the third case (when the root exists), we put all the probability on the root $a'$, and the dot product reduces to:

$$c \cdot (p_t \odot \pi(y)) = c(a') p_t(a') \pi(a', y) = 0$$

since $c(a')$ is zero (and note that we defined the inner product $c \cdot (p_t \odot \pi(y))$ to generalize to the function $c$). $\square$

### C.1.2 Quantile Calibration

Next, we are interested in a generalization of the above setting, in which the output of the algorithm is an entire quantile function.

**Setup** We are given as input a quantile function $Q_t(a)$ (that could be different at each time step) that targets a continuous outcome $y_t \in \mathbb{R}$ that is bounded by values $y_{\min}, y_{\max}$. Our goal is to choose at each time step a recalibrator function $R_t : [0, 1] \to [0, 1]$ such that the corrected quantile function $Q_t' = Q_t \circ R_t$ represents valid quantiles within an error tolerance of $\epsilon > 0$ for all $\beta$. In other words, we want

$$\lim_{T \to \infty} \left( \frac{1}{T} \sum_{t=1}^{T} \mathbf{I}\{y_t \leq Q'(\beta)\} - \beta \right) \leq \epsilon \text{ for all } \beta > 0.$$

**Deterministic Algorithm** Without loss of generality, we can define our action space to consist of quantile functions $Q_t$. We define our **payoffs** as follows. Let $e_t(a) = \mathbf{I}\{y_t \leq Q(a)\}$. Let $\pi(Q, a, y) = e_t(a) - a$, which is a vector indexed by $a \in [0, 1]$. We want the average set of payoffs to approach a ball of size $\epsilon$ centered at the origin. Note that this yields a valid quantile estimate as defined above.

In order to achieve this goal using approachability, we need to devise a half-space oracle that will output at each time step a $Q_t$ such that for any vector of historical payoffs $c \in \mathbb{R}^m$, we have $c \cdot \pi_t(Q, y) \leq 0$ for all $y$, where $\pi_t(Q, y)$ is the vector of payoffs indexed by $A$ and $\cdot$ denotes the inner product.

We construct the following **half-space oracle** for this goal. Suppose we are given a vector of historical payoffs $c$. We run a root finding-algorithm to find at least one root (or the absence of any root).

- If $c \geq 0$, let $Q(a) = y_{\min}$ for all $a$, and output $Q$.

- If $c \leq 0$, let $Q(a) = y_{\max}$ for all $a$, and output $Q$.

- Otherwise, there is a zero $a'$ such that $c(a') = 0$. Let $A_1$ be the integral under the curve of $c$ for $a \leq a'$ and let $A_2$ be the integral under the curve for $a \geq a'$.

  - If $A_2 - A_1 \leq 0$, let $Q(a) = y_{\min}$ for $a \leq a'$ and $Q(a) = y_{\max}$ for $a \leq a'$.
  - If $A_1 + A_2 \leq 0$ let $Q(a) = y_{\max}$ for all $a$, and output $Q$.
  - If $A_1 + A_2 \geq 0$ let $Q(a) = y_{\min}$ for all $a$, and output $Q$.

  is positive and the area under the curve for $a \geq a'$ is negative, let $Q(a) = y_{\min}$ for $a \leq a'$ and $Q(a) = y_{\max}$ for $a \leq a'$.

**Lemma C.3.** *The above algorithm is a deterministic half-space oracle for a ball of radius $\epsilon$. Its runtime is equal to that of the root-finding oracle.*

*Proof.* First, note that the above three cases in the algorithm are exhaustive. Next, note that when $Q(a) = y_{\max}$, then $\pi(Q, a, y) \geq 0$ for all $y$. When $Q(a) = y_{\min}$, then $\pi(Q, a, y) \leq 0$ for all $y$. Thus, in each of the first two cases we have $c \cdot \pi_t(Q, y) \leq 0$.

In the third setting, first subcase, by construction we have exactly $c \cdot \pi_t(Q, y) \leq A_2 - A_1 \leq 0$. The last two subcases follow similarly. Note that computationally, we can distinguish among all these subcases, by performing an inner product with $c$ and observing the result. The computational cost of the algorithm is that of running the root finding algorithm and the inner product. $\square$

Note that this algorithm is not practical, as it produces $Q$'s that are step functions. It is meant to illustrate the existence of a polynomial algorithm in our framework. In practice, one would use our optimization-based solution.

## C.2 Variations of Distribution Calibration

Next, we are going to derive half-space oracles for versions of distribution calibration, both the original version of distribution calibration, as well as more restricted and tractable versions, including decision calibration and our novel formulation of moment-based calibration.

### C.2.1 Distribution Calibration via Discretization

Our overall strategy for constructing half-space oracles will be inspired by Kakade & Foster (2008). We first provide a summary of their approach that closely follows their exposition.

Specifically, we will define a discretization $V$ of the space of forecasts. For example, the set V could consist of probability distributions which are specified to a finite number of digits of precision. Essentially, our quasi-randomized approach will output a distribution over a small number of similar elements of $V$, and our deterministic approach will play a fixed point of this distribution.

More specifically, we will assume that the forecasts live in a compact set $\Delta$ (this assumption will have to be verified for each definition of calibration). We then define a triangulation of $\Delta$, i.e., a partition into a set of simplices such that any two simplices intersect in either a common face, common vertex, or not at all. Let $V$ be the vertex set of this triangulation. Note that any point $p$ lies in some simplex in this triangulation, and, slightly abusing notation, let $V(p)$ be the set of corners for this simplex. We are going to produce a randomized output over these corners.

To formalize this, associate a test function $w_v(p)$ with each $v \in V$ as follows. Each distribution $p$ can be uniquely written as a weighted average of its neighboring vertices, $V(p)$. For $v \in V(p)$, let us define the test functions $w_v(p)$ to be these linear weights, so they are uniquely defined by the linear equation $p = \sum_{v \in V(p)} w_v(p)v$.

We also define the discretization to be sufficiently small: given a target precision $\epsilon > 0$ we define the discretization such that for all $f_1, f_2$ in the same simplex we have $||f_1 - f_2|| < \epsilon$.

**Deterministic and Quasi-Deterministic Calibration**   We use the following results from Kakade & Foster (2008). Let $\mu_T(v) = \frac{1}{T}\sum_{t=1}^{T} w_v(f_t)(y_t - f_t)$. For $v \in V$, define $\rho_T(v)$, a function which updates $v$ using the calibration error $\mu_T(v)$: $\rho_T(v) = v + \mu_T(v)$. We extend this function to arbitrary $p$ by interpolating between the vertices of the cell that contains $p$: $\rho_T(p) = p + \sum_{v \in V} w_v(p)\mu_T(v)$.

Consider the following "play the fixed point" algorithm defined by Kakade & Foster (2008). At time $T = 1$, we set $\mu_0(v) = 0$ for all $v \in V$. Then at each future time $T$, compute a fixed point of $\rho_{T-1}$ and forecast this fixed point.

We will use the following facts

**Lemma C.4.** *For all $T$, a fixed point of $\rho_T$ exists and the forecast $f_T$ at time $T$ satisfies* $\sum_{v \in V} w_v(f_T)\mu_{T-1}(v) = 0$.

**Lemma C.5.** *The above deterministic algorithm is weakly calibrated in the sense that* $\frac{1}{T}\lim \to \infty \sum_{t=1}^{T} w(f_t)(y_t - f_t) \to 0$ *in the $\ell_\infty$ norm for any continuous function $w$.*

Consider now the following quasi-deterministic version of the above algorithm. We start with a forecasting procedure that is weekly calibrated. At time $t$, that procedure outputs a forecast $f_t$. We define a quasi-deterministic forecast by outputting the vertices $v \in V(f_t)$ of the simplex containing $f_t$, each with probability $w_v(f_t)$. Recall that we have chosen the discretization such that $||f_1 - f_2|| < \epsilon$.

**Lemma C.6.** *The limit of the calibration error of the above algorithm as $T \to \infty$ is at most $\epsilon$.*

Next, we will use these tools to establish half-space oracles for our framework.

### C.2.2   Moment-Based Distribution Calibration

**Setup**   In the setting of moment-based calibration, we are trying to forecast at each time step $t$ a continuous label $y_t \in \mathbb{R}$ and we assume that $|y_t| \leq B$ is bounded by $B > 0$. Our goal is to choose at each step $t$ a prediction $\mu_t, \sigma_t$ such that on average, our of all the times when we predicted $\mu_t, \sigma_t$ the mean and the variance of the $y_t$ are also approximately equal $\mu_t, \sigma_t$.

**Randomized Algorithm**   We define our **action set** as $A = \{(\mu_i, \sigma_j)\}$ for $i, j$ in a grid with $M$ ticks along each dimension). The grid represents our set if simplexes and the ticks on the grid are the vertices $V$. We will output a probability $p$ over A and sample a forecast $f_t$ from $p$. We define a set of **payoffs** as $\pi(\mu, \sigma, y) = (\mu - y, \sigma_- y^2])$; thus $\pi$ has dimension $2M^2$. We want the average set of payoffs to approach a ball of size $\epsilon$ centered at the origin. Note that this yields a valid quantile estimate as defined above.

In order to achieve this goal using approachability, we need to devise a half-space oracle that will output at each time step a distribution $p_t$ over $A$ such that for any vector of historical payoffs $c \in \mathbb{R}^m$, we have $c^\top(p_t \odot \pi_t(y)) \leq 0$ for all $y$, where $\pi_t(y)$ is a vector of payoffs indexed by $A$.

We construct the following **half-space oracle** for this goal. Suppose we are given a vector of historical payoffs $c$. By the above fixed point lemma, there must exist a fixed point $f_t$ such that $\sum_{v \in V} w_v(f_T)\mu_{T-1}(v) = 0$. We can find the cell where $f_t$ lives by enumerating the $M^2$ cells. Then we set $p_t$, our probability over the $A$ to be zero everywhere except at $V(f_t)$ and equal to $w_v(f_t)$ everywhere else.

**Lemma C.7.** *Let $\epsilon > 1/M$. Then the above algorithm is a half-space oracle for a ball of radius $\epsilon$. Furthermore, the algorithm runs in $\epsilon^2$ time and $O(1/\epsilon^2)$ space.*

*Proof.* First, note that our set of possible forecasts is compact, which guarantees that the fixed point exists.

Next, we prove that the property of the half-space oracle is satisfied. If the fixed point is in the interior of a cell, note that:

$$c_t^\top(p_t \odot \pi(y)) = \sum_{v \in V} w_v(f_t)\mu_{t-1}(v) = 0$$

because $c_t$ is defined to be previous vector of average payoffs, which is precisely $\mu_{t-1}$.

Note that there may be edge cases, where the fixed point is at the edge of the grid. Then we simply have to enumerate the following edge cases. Suppose that $c_{\sigma,i,j=0} \geq 0$ for all $i$ (it's fully positive on the left edge of the grid). Then you can find an optimal strategy over $\mu$ as in a 1d problem over $(i, j = 0)$. Suppose that $c_{\sigma,i,j=M} \leq 0$ for all $i$ (it's fully positive on the right edge) Then you can find an optimal strategy over $\mu$ as in a 1d problem over $(i, j = M)$. Suppose that $c_{\mu,i=0,j} \geq 0$ for all $j$ (it's fully positive on the top edge of the grid) Then you can find an optimal strategy over $\sigma$ as in a 1d problem over $(i = 0, j)$. Suppose that $c_{\mu,i=M,j} \leq 0$ for all $j$ (it's fully positive on the bottom edge) Then you can find an optimal strategy over $\sigma$ as in a 1d problem over $(i = M, j)$.

Next, we explain the above time and space complexity of the algorithm. First, the space complexity is $O(M^2) = O(1/\epsilon^2)$ because $\epsilon > 1/M$. The time complexity is $O(M^2) = O(\epsilon^2)$ because we simply enumerate all the cells. □

**Deterministic Algorithm**   The above procedure naturally leads to a natural randomized algorithm, where we simply forecast the fixed point. Computationally, implementing this algorithm requires an oracle for Brouwer's fixed-point problem (in general, this is PPAD-hard).

### C.2.3   General Distribution Calibration

We can extend the above approach to more general settings, where we want to main a general notion of distribution calibration. In other words, out of the times when we predict $f_t$, we want the distribution of $y$ to look like $f_t$.

**Setup**   We are trying to forecast at each time step $t$ a continuous label $y_t \in \mathbb{R}$ and we assume that $|y_t| \leq B$ is bounded by $B > 0$. We are going to assume a certain discretization for $y$ into a partition of $M$ intervals of its domain. Our goal is to choose at each step $t$ a forecast $f_t$ such that on average, our of all the times when we predicted $y_t$ we have $y_t \approx f_t$: this can be roughly viewed as matching probability mass functions.

**Randomized Algorithm**   We define our **action set** $A$ as a discretization of the set of distributions $f_t$. This can represent a discretization of the range of $f_t$ into a grid of $N$ points. The grid represents our set if simplexes and the ticks on the grid are the vertices $V$. We will output a probability $p$ over A and sample a forecast $f_t$ from $p$. We define a set of **payoffs** as $\pi(f, y) = f - y$; thus $\pi$ has dimension $O(M^N)$. We want the average set of payoffs to approach a ball of size $\epsilon$ centered at the origin. Note that this yields a valid quantile estimate as defined above.

Again, we need to devise a half-space oracle that will output at each time step a distribution $p_t$ over $A$ such that for any vector of historical payoffs $c \in \mathbb{R}^m$, we have $c^\top(p_t \odot \pi_t(y)) \leq 0$ for all $y$, where $\pi_t(y)$ is a vector of payoffs indexed by $A$.

We construct the following **half-space oracle** for this goal. Suppose we are given a vector of historical payoffs $c$. By the above fixed point lemma, there must exist a fixed point $f_t$ such that $\sum_{v \in V} w_v(f_T)\mu_{T-1}(v) = 0$. We can find the cell where $f_t$ lives by enumerating the $M^N$ cells. Then we set $p_t$, our probability over the $A$ to be zero everywhere except at $V(f_t)$ and equal to $w_v(f_t)$ everywhere else.

**Lemma C.8.** *The above algorithm is a half-space oracle for a ball of radius $\epsilon$.*

*Proof.* First, note that our set of possible forecasts is compact, which guarantees that the fixed point exists.

Next, we prove that the property of the half-space oracle is satisfied. If the fixed point is in the interior of a cell, note that:

$$c_t^\top(p_t \odot \pi(y)) = \sum_{v \in V} w_v(f_t)\mu_{t-1}(v) = 0$$

because $c_t$ is defined to be previous vector of average payoffs, which is precisely $\mu_{t-1}$. Edge cases would be handled similarly to the earlier proof. □

Note that in most cases, this algorithm would not be computationally tractable. However, that is expected: the problem is in general PPAD-hard. This question of computational tractability is what motivated our previous definition of moment-based calibration.

**Deterministic Algorithm**   The above procedure naturally leads to a natural randomized algorithm, where we simply forecast the fixed point. Computationally, implementing this algorithm, requires access to an oracle's for Brower's fixed-point problem (in general, this is PPAD-hard).

## D   Low Regret Relative to Baseline Classifiers

Here, we show that a calibrated forecaster also has small regret relative to any bounded proper loss if we use a certain construction that combines our algorithm with a baseline forecaster.

### D.1   Recalibration Construction

**Setup**   We start with an online forecaster $F$ that outputs uncalibrated forecasts $p_t^F$ at each step; these forecasts are fed into a *recalibrator* such that the resulting forecasts $p_t$ are calibrated and have low regret relative to the baseline forecasts $p_t^F$.

Formally, at every step $t = 1, 2, ...$ we have:
1: Forecaster $F$ predicts $p_t^F$.
2: A recalibration algorithm produces a calibrated forecast $p_t$ based on $p_t^F$.
3: Nature reveals label $y_t$
4: Based on $x_t, y_t$, we update the recalibration algorithm and optionally update $H$.

**Notation**   We will reuse some of the previously introduced construction based on the work of (Kakade & Foster, 2008). Specifically, recall that we define a discretization $V$ of the space of forecasts. We assume that the forecasts live in a compact set $\Delta$ and we define a triangulation of $\Delta$, i.e., a partition into a set of simplices such that any two simplices intersect in either a common face, common vertex, or not at all. Let $V$ be the vertex set of this triangulation, and let $V(p)$ be the set of corners for this simplex.

Note that each distribution $p$ can be uniquely written as a weighted average of its neighboring vertices, $V(p)$. For $v \in V(p)$, we define the test functions $w_v(p)$ to be these linear weights, so they are uniquely defined by the linear equation $p = \sum_{v \in V(p)} w_v(p)v$. We also define the discretization to be sufficiently small: given a target precision $\epsilon > 0$ we define the discretization such that for all $f_1, f_2$ in the same simplex we have $||f_1 - f_2|| < \epsilon$.

### D.2   Recalibration Algorithm

We are going to define a general meta-algorithm that follows a construction in which we run multiple instances of our calibrated forecasting algorithms over the inputs of $F$.

More formally, we take the aforementioned partition of the space of forecasts of $\Delta$ of $F$ and we associate each simplex with an instance of our calibration algorithm $F^{\text{cal}}$ (using the same $\Delta$ and discretization $V$). In order to compute $p_t^F$, we invoke the subroutine $F_j^{\text{cal}}$ associated with simplex $I_j$ containing $p_t^F$ (with ties broken arbitrarily). After observing $y_t$, we pass it to $F_j^{\text{cal}}$.

The resulting procedure produces valid calibrated estimates because each $F_j^{\text{cal}}$ is a calibrated subroutine. More importantly the new forecasts do not decrease the predictive performance of $F$, as measured by a proper loss $\ell$. In the remainder of this section, we establish these facts formally.

### D.3   Theoretical Analysis

**Notation**   Our task is to produce calibrated forecasts. Intuitively, we say that a forecast $F_t$ is calibrated if for every $y' \in \mathcal{Y}$, the probability $F_t(y')$ on average matches the frequency of the event $\{y = y'\}$. We formalize

this by introducing the ratio

$$\rho_T(p) = \frac{\sum_{t=1}^{T} y_t \cdot \mathbb{1}_{p_t=p}}{\sum_{t=1}^{T} \mathbb{1}_{p_t=p}} \tag{20}$$

Intuitively, we want $\rho_T(p) \to p$, a.s. as $T \to \infty$ for all $y$. In other words, out of the times when the predicted probability for $y_t$ is $p$, the average $y_t$ look like $p$.

The quality of probabilistic forecasts is evaluated using *proper* losses $\ell$. Formally, a loss $\ell(y,p)$ is proper if $p \in \arg\min_{q \in \mathcal{P}} \mathbb{E}_{y \sim (p)} \ell(y,q) \; \forall p \in \mathcal{P}$. An example in binary classification is the log-loss $\ell_{\log}(y,p) = y \log(p) + (1-y) \log(1-p)$. We will assume that the loss is bounded by $B > 0$ .

We measure calibration a calibration error $C_T$. Our algorithms will output discretized probabilities; hence we define the error relative to a set of possible predictions $V$

$$C_T = \sum_{p \in V} |\rho_T(p) - p| \left( \frac{1}{T} \sum_{t=1}^{T} \mathbb{1}_{\{p_t=p\}} \right). \tag{21}$$

### D.3.1  A Helper Lemma

In order to establish the correctness of our recalibration procedure, we need to start with a helper lemma. This lemma shows that if forecasts are calibrated, then they have small internal regret.

**Lemma D.1.** *If $\ell$ is a bounded proper loss, then an $(\epsilon, \ell_1)$-calibrated $F^{cal}$ a.s. has a small internal regret w.r.t. $\ell$ and satisfies uniformly over time $T$ the bound*

$$R_T^{int} = \max_{ij} \sum_{t=1}^{T} \mathbb{1}_{p_t=p_i} \left( \ell(y_t, p_i) - \ell(y_t, p_j) \right) \leq 2B(R_T + \epsilon). \tag{22}$$

*Proof.* Let $T$ be fixed for the rest of this proof. Let $\mathbb{1}_{ti} = \mathbb{1}_{p_t=p_i}$ be the indicator of $F^{cal}$ outputting prediction $p_i$ at time $t$, let $T_i = \sum_{t=1}^{T} \mathbb{1}_{ti}$ denote the number of time $i/N$ was predicted, and let

$$R_{T,ij}^{int} = \sum_{t=1}^{T} \mathbb{1}_{ti} \left( \ell(y_t, p_i) - \ell(y_t, p_j) \right)$$

denote the gain (measured using the proper loss $\ell$) from retrospectively switching all the plays of action $i$ to $j$. This value forms the basis of the definition of internal regret.

Let $T(i,y) = \sum_{t=1}^{T} \mathbb{1}_{ti} \mathbb{1}\{y_t = y\}$ denote the total number of $p_i$ forecasts at times when $y_t = y$. Observe that we have

$$T(i,y) = \sum_{t=1}^{T} \mathbb{1}_{ti} \mathbb{1}\{y_t = y\} = \frac{\sum_{t=1}^{T} \mathbb{1}_{ti} \mathbb{1}\{y_t = y\}}{T_i} T_i = \frac{\sum_{t=1}^{T} \mathbb{1}_{ti} \mathbb{1}\{y_t = y\}}{\sum_{t=1}^{T} \mathbb{1}_{ti}} T_i$$

$$= q(i,y) T_i + T_i \left( \frac{\sum_{t=1}^{T} \mathbb{1}_{ti} \mathbb{1}\{y_t = y\}}{\sum_{t=1}^{T} \mathbb{1}_{ti}} - q(i,y) \right)$$

$$= q(i,y) T_i + T_i \left( \rho_T(p_i) - p_i \right),$$

where $q(i,y) = p_i(y)$. The last equality follows using some simple algebra after adding and subtracting one inside the parentheses in the second term.

We now use this expression to bound $R_{T,ij}^{\text{int}}$:

$$
\begin{aligned}
R_{T,ij}^{\text{int}} &= \sum_{t=1}^{T} \mathbb{I}_{ti} \left( \ell(y_t, p_i) - \ell(y_t, p_j) \right) \\
&= \sum_y T(i,y) \left( \ell(y, p_i) - \ell(y, p_j) \right) \\
&\leq \sum_y q(i,y) T_i \left( \ell(y, p_i) - \ell(y, p_j) \right) + BT_i \left| \rho_T(p_i) - p_i \right| \\
&\leq BT_i \left| \rho_T(p_i) - p_i \right|,
\end{aligned}
$$

where in the first inequality, we used $\ell(y, p_i) - \ell(y, p_j) \leq \ell(y, p_i) \leq B$, and in the second inequality we used the fact that $\ell$ is a proper loss.

Since internal regret equals $R_T^{\text{int}} = \max_{i,j} R_{T,ij}^{\text{int}}$, we have

$$
R_T^{\text{int}} \leq \sum_{i=1}^{N} \max_j R_{T,ij}^{\text{int}} \leq 2B \sum_{i=0}^{N} T_i \left| \rho(i/N) - p_i \right| \leq 2B(R_T + \epsilon).
$$

$\square$

### D.4  Recalibrated forecasts have low regret relative to uncalibrated forecasts

Next, we use the above result to prove that the forecasts recalibrated using the above construction have low regret relative to the baseline uncalibrated forecasts.

**Lemma D.2** (Recalibration preserves accuracy). *Let $\ell$ be a bounded proper loss such that $\ell(y_t, p) \leq \ell(y_t, p_j) + B\epsilon$ whenever $||p - p_j|| \leq \epsilon$. Then the recalibrated $p_t$ a.s. have vanishing $\ell$-loss regret relative to $p_t^F$ and we have uniformly:*

$$
\frac{1}{T} \sum_{t=1}^{T} \ell(y_t, p_t) - \frac{1}{T} \sum_{t=1}^{T} \ell(y_t, p_t^F) < \frac{B}{\epsilon} \sum_{j=1}^{M} \frac{T_j}{T} R_{T_j} + 3B\epsilon. \tag{23}
$$

*Proof.* By the previous lemma, we know that an algorithm whose calibration error is bounded by $R_T = o(1)$ also minimizes internal regret at a rate of $2BR_T$, and thus external regret at a rate of $2BR_T/\epsilon$.

Next, let us use $\mathbb{I}_{j,t}$ to indicate that $F_j^{\text{cal}}$ was called at time $t$. We establish our main claim as follows:

$$
\begin{aligned}
&\frac{1}{T} \sum_{t=1}^{T} \ell(y_t, p_t) - \frac{1}{T} \sum_{t=1}^{T} \ell(y_t, p_t^F) \\
&= \frac{1}{T} \sum_{t=1}^{T} \left( \sum_{j=1}^{M} \left( \ell(y_t, p_t) - \ell(y_t, p_t^F) \right) \mathbb{I}_{j,t} \right) \\
&< \frac{1}{T} \sum_{t=1}^{T} \left( \sum_{j=1}^{M} \left( \ell(y_t, p_t) - \ell(y_t, p_j) \right) \mathbb{I}_{j,t} + B\epsilon \right) \\
&\leq \frac{1}{\epsilon} B \sum_{j=1}^{M} \frac{T_j}{T} R_{T_j} + 3B\epsilon,
\end{aligned}
$$

where $R_{T_j}$ is a bound on the calibration error of $F_j^{\text{cal}}$ after $T_j$ plays.

In the first two inequality, we use our assumption on the loss $\ell$. The last inequality follows because $F_j^{\text{cal}}$ minimizes external regret w.r.t. the constant action $p_j$ at a rate of $BR_{T_j}/\epsilon$. $\square$

### D.5 Proving that calibration holds under any norm

We want to also give a proof that the recalibration construction described above yields calibrated forecasts.

**Lemma D.3.** *If each $F_j^{cal}$ is $(\epsilon, \ell_p)$-calibrated, then the combined algorithm is also $(\epsilon, \ell_p)$-calibrated and the following bound holds uniformly over $T$:*

$$C_T \le \sum_{j=1}^{M} \frac{T_j}{T} R_{T_j} + \epsilon. \tag{24}$$

*Proof.* Let $M = |V|$. Let $\mathbb{I}_i^{(j)} = \sum_{t=1}^{T} \mathbb{I}_{t,i}^{(j)}$ where $\mathbb{I}_{t,i}^{(j)} = \mathbb{I}\{p_t = p_j \cap p_t^F = p_j\}$ and note that $\sum_{t=1}^{T} \mathbb{I}_{t,i} = \sum_{j=1}^{M} \mathbb{I}_i^{(j)}$. Let also $\rho_T^{(j)}(p_i) = \frac{\sum_{t=1}^{T} \mathbb{I}_{t,i}^{(j)} y_t}{\sum_{t=1}^{T} \mathbb{I}_{t,i}^{(j)}}$. We may write

$$C_{T,i} = \frac{\sum_{t=1}^{T} \mathbb{I}_{t,i}}{T} |\rho_T(p_i) - p_i| = \frac{\sum_{j=1}^{M} \mathbb{I}_i^{(j)}}{T} \left| \sum_{j=1}^{M} \frac{\sum_{t=1}^{T} \mathbb{I}_{t,i}^{(j)} y_t}{\sum_{j=1}^{M} \mathbb{I}_i^{(j)}} - p_i \right|$$

$$= \frac{\sum_{j=1}^{M} \mathbb{I}_i^{(j)}}{T} \left| \sum_{j=1}^{M} \frac{\mathbb{I}_i^{(j)} \rho_T^{(j)}(p_i)}{\sum_{j=1}^{M} \mathbb{I}_i^{(j)}} - p_i \right| \le \sum_{j=1}^{M} \frac{\mathbb{I}_i^{(j)}}{T} \left| \rho_T^{(j)}(p_i) - p_i \right| = \sum_{j=1}^{M} \frac{T_j}{T} C_{T,i}^{(j)},$$

where $C_{T,i}^{(j)} = \left| \rho_T^{(j)}(p_i) - p_i \right| \left( \frac{1}{T_j} \sum_{t=1}^{T} \mathbb{I}_{t,i}^{(j)} \right)$ and in the last line we used Jensen's inequality. Plugging in this bound in the definition of $C_T$, we find that

$$C_T = \sum_{i=1}^{N} C_{T,i} \le \sum_{j=1}^{M} \sum_{i=1}^{N} \frac{T_j}{T} C_{T,i}^{(j)} \le \sum_{j=1}^{M} \frac{T_j}{T} R_{T_j} + \epsilon,$$

Since each $R_{T_j} \to 0$, the full procedure will be $\epsilon$-calibrated. $\qquad\square$

Recall that $R_T$ denotes the rate of convergence of the calibration error $C_T$. For most online calibration subroutines $F^{cal}$, $R_T \le f(\epsilon)/\sqrt{T}$ for some $f(\epsilon)$. In such cases, we can further bound the calibration error in the above lemma as

$$\sum_{j=1}^{M} \frac{T_j}{T} R_{T_j} \le \sum_{j=1}^{M} \frac{\sqrt{T_j} f(\epsilon)}{T} \le \frac{f(\epsilon)}{\sqrt{\epsilon T}}.$$

In the second inequality, we set the $T_j$ to be equal. Thus, our recalibration procedure introduces an overhead of $\frac{1}{\sqrt{\epsilon}}$ in the convergence rate of the calibration error $C_T$ and of the regret relative to a baseline forecaster in the earlier lemma.

## E    Applications: Decision-Making

Next, we complement our results with a formal characterization of some benefits of calibration. Consider a decision-task where we wish to estimate a value function $v : \mathcal{Y} \times \mathcal{A} \times \mathcal{X} \to \mathbb{R}$ over a set of outcomes $\mathcal{Y}$, actions $\mathcal{A}$, and features $\mathcal{X}$. Note that the function $v$ could be a loss $\ell(y, a, x)$ that quantifies the error of an action $a \in \mathcal{A}$ in a state $x \in \mathcal{X}$ given outcome $y \in \mathcal{Y}$.

We assume that given $x$, the agent chooses an action $a(x)$ according to a decision-making process. This could be an action $a(x) = \arg\min_a \mathbb{E}_{y \sim H(x)}[\ell(y, a, x)]$ that minimizes an expected loss according to the probabilities given by the forecast. The agent then relies on a predictive model $H$ of $y$ to estimate the future values $v(y, a, x)$ for the decision $a(x)$ :

$$v(x) = \mathbb{E}_{y \sim H(x)}[v(y, a(x), x)]. \tag{25}$$

We study $v(y, a, x)$ that are monotonically non-increasing or non-decreasing in $y$. Examples include linear utilities $u(a, x) \cdot y + c(a, x)$ or their monotone transformations.

**Expectations Under Calibrated Models**   If $H$ was a perfect predictive model, we could estimate expected values of outcomes perfectly. In practice, inaccurate models can yield imperfect decisions. Surprisingly, our analysis shows that in many cases, calibration (a much weaker condition that having a perfectly specified model $H$) is sufficient to correctly estimate the value of various outcomes.

Surprisingly, our guarantees can be obtained with a weak condition—quantile calibration. Additional requirements are the non-negativity and monotonicity of $v$. Our result is a concentration inequality that shows that estimates of $v$ are unlikely to exceed the true $v$ on average  (Zhao et al., 2020).

**Theorem E.1.** *Let $M$ be a quantile calibrated model as in and let $v(y, a, x)$ be a monotonic value function. Then for any sequence $(x_t, y_t)_{t=1}^T$ and $r > 1$, we have:*

$$\lim_{T \to \infty} \frac{1}{T} \sum_{t=1}^{T} \mathbb{I}\left[v(y_t, a(x_t), x_t) \geq rv(x_t))\right] \leq 1/r \tag{26}$$

*Proof.* Recall that $M(x)$ is a distribution over $\mathcal{Y}$, with a density $p_x$, a quantile function $Q_x$, and a cdf $F_x$. Note that for any $x$ and $s \in (0, 1)$ and $y' \leq F_x^{-1}(1 - s)$ we have:

$$v(x) = \int v(x, y, a(x)) q_x(y) dy$$

$$\geq \int_{y \geq y'} v(x, y, a(x)) q_x(y) dy$$

$$\geq v(x, y', a(x)) \int_{y \geq y'} q_x(y) dy$$

$$\geq sv(x, y', a(x))$$

The above logic implies that whenever $v(x) \leq sv(x, y, a)$, we have $y \geq F_x^{-1}(1 - s)$ or $F_x(y) \geq (1 - s)$. Thus, we have for all $t$,

$$\mathbb{I}\{v(x_t) \leq sv(x_t, y_t, a_t)\} \leq \mathbb{I}\{F_{x_t}(y_t) \geq (1 - s)\}.$$

Therefore, we can write

$$\frac{1}{T} \sum_{t=1}^{T} \mathbb{I}\{v(x_t) \leq sv(x_t, y_t, a_t)\} \leq \frac{1}{T} \sum_{t=1}^{T} \mathbb{I}\{F_{x_t}(y_t) \geq (1 - s)\} = s + o(T),$$

where the last equality follows because $M$ is calibrated. Therefore, the claim holds in the limit as $T \to \infty$ for $r = 1/s$. The argument is similar if $v$ is monotonically non-increasing. In that case, we can show that whenever $y' > F_x^{-1}(s)$, we have $v(x) \geq sv(x, y', a(x))$. Thus, whenever $v(x) \leq sv(x, y, a)$, we have $y \leq F_x^{-1}(s)$ or $F_x(y) \leq s$. Because, $F_x$ is calibrated, we again have that

$$\frac{1}{T} \sum_{t=1}^{T} \mathbb{I}\{v(x_t) \leq sv(x_t, y_t, a_t)\} \leq \sum_{t=1}^{T} \mathbb{I}\{F_{x_t}(y_t) < s\} = s + o(T),$$

and the claim holds with $r = 1/s$. $\qquad\qquad\qquad\qquad\qquad\qquad\qquad\qquad\qquad\qquad\qquad\square$

Note that this statement represents an extension of Markov inequality. Note also that this implies the same result for a distribution calibrated model, since distribution calibration implies quantile calibration.

## F   Additional Experimental Details

Experiments were run on CPU on a MacBook Pro with an M2 chip and 64 GB of memory. The results in our experiments were generated using 6 hours of compute.

