# OpenReview forum: "Calibrated Probabilistic Forecasts for Arbitrary Sequences"
_TMLR — Accepted by TMLR_

### Review · Reviewer_dQat · 2024-12-19

**Summary Of Contributions:**

- The author(s) proposed a game-theoretical framework to study calibration in online learning settings.
- From the framework, the author showed an algorithm can achieve calibration when having access to an oracle.
- For some notions of calibration, such oracle is accessible, for those not, the author(s) proposed a gradient-based method (ORCA) to approximate.

**Audience:**

Yes

**Claims And Evidence:**

Yes

**Requested Changes:**

- Could the author comment on how the approximation made from eq (8) to eq (9) might impact the calibration if not showing a result?
- One thing I am a bit confused about, which might be due to the lack of background, is how the feature $x_t$ enters the theory. It was carried in the pay-off function and as the authors noted so long as the pay-off takes the form that $x$ only in the condition function their theory would carry through. I would imaging a forecaster deciding their $p_t$ partly based on $x_t$, I wonder how this type of dependency is captured in the framework. E.g., as the authors noted in binary distribution, calibration one wants $P(y_t=1|p_t\approx p)\approx p$ to happen often. I wonder is there a notion that this also condition on $x_t$? Can something like this be written as the general pay-off the authors worked on?

**Strengths And Weaknesses:**

I am not familiar with literature in game theory but to me,
Strengths
- The game theory formalism fits many notions of calibration that can be written as pay-offs and is insightful.
- The proposed gradient method, despite being an approximation, seems perform well, at least not worse than existing methods.
- The paper is generally clear, and an enjoyable read. Thank you!!

Weakness:
- The behavior of Algorithm 2 does not seem well understood. I suspect there is a way to quantify loss due to the approximation from equation (8) to (9) and having a residual term in pay-off.

---

> ### Author Response · Authors · 2025-01-10
>
> Thank you for your feedback! In particular, your suggestion about adding details for the approximation from equation (8) to (9) was helpful. See our responses to your questions below.
>
> > Could the author comment on how the approximation made from eq (8) to eq (9) might impact the calibration if not showing a result?
>
> The approximation for the inner maximization weakens Nature, which may lead us to underestimate the cost of the worst-case outcome. However, the impact of this approximation can be bounded. If the component distributions $q_k$ are each supported on sets of diameter $d$, and the union of their supports forms an $\epsilon$-net on $\mathcal{Y}$, then the approximating family can guarantee that the outcome is within distance $d+\epsilon$ of the worst-case outcome. Thus, if the payoff is $L$-Lipschitz in $y$ then the impact of this approximation is bounded by $0 \leq  A - \widehat{A}  \leq (d+\epsilon)L \cdot \|\| \overline \pi_{t-1} \|\|$, where $A := \max_{q \in \Delta(\mathcal{Y})} ~ \underset{y \sim q}{\mathbb{E}} \left[\langle \overline\pi_{t-1}, \pi(x_t, p, y)  \rangle \right]$ is the exact expression in equation (8) and $\widehat{A} := \max_{\phi \in \Delta_{k-1}} ~ \sum_{k=1}^K \phi_k \underset{y \sim q_k}{\mathbb{E}}\left[\langle \overline\pi_{t-1}, \pi(x_t, p, y)  \rangle \right]$ is the approximate expression in equation (9).
>
> Conversely, the approximation for the outer maximization weakens Forecaster, which may lead us to overlook the best forecast. One could make similar arguments as above to bound the impact of this approximation, or simply note that this approximation does not weaken Nature, so while this approximation can prevent us from realizing the optimal worst-case guarantee, the validity of the guarantee we *do* identify is preserved. We will include this discussion in the updated manuscript.
>
> > One thing I am a bit confused about, which might be due to the lack of background, is how the feature $x_t$ enters the theory. It was carried in the pay-off function and as the authors noted so long as the pay-off takes the for that $x$ only in the condition function their theory would carry through. I would imagining a forecaster deciding their $p_t$ partly based on $x_t$, I wonder how this type of dependency is captured in the framework. E.g., as the authors noted in binary distribution, calibration one wants $P(y_t = 1 | p_t \approx p) \approx p$ to happen often. I wonder is there a notion that this also condition on $x_t$? Can something like this be written as the general pay-off the authors worked on?
>
> Great question! The features $x_t$ can naturally be incorporated into the payoff and influence the forecasts. For a few examples, $x_t$ is used in the payoffs for local calibration, conditional calibration, and no-regret recalibration. For conditional calibration, this looks like $P(y_t = 1 | x_t \in S_k, p_t \approx p) \approx p, \forall k \in [1, …, K]$ where $S_1, \dots, S_K \subset \mathcal{X}$ are the subsets of the feature space on which we want calibration. For no-regret recalibration, the features $x_t$ are critical to incorporate into the payoff to achieve the no-regret guarantee, and in practice this drastically affects the generated forecasts. Please feel free to follow up if you have further questions here!

---

> > ### Comment · Reviewer_dQat · 2025-01-10
> >
> > Thank you for the response! Both answered my question and the the discussion on boundedness of the approximation error is particularly helpful for me. I look forward for an updated version!

---

### Review · Reviewer_XmLe · 2024-12-25

**Summary Of Contributions:**

This work proposes a unifying framework for miscalibration in probabilistic forecasting. The authors model miscalibration as the average payoff of a zero-sum game between a Forecaster and Nature, where the Forecaster's goal is to drive the average payoff to 0. They show that, under mild conditions, miscalibration decreases at a rate proportional to the inverse square root of the number of rounds. They extend this framework to enable recalibrating forecasters with calibration and no-regret guarantees.

The optimal forecast is found through a minimax problem where the Forecaster minimizes the worst-case forecast error induced by Nature’s actions. As this problem is computationally hard, the authors propose a general gradient-based method that solves a relaxation of this problem and also derive specialized algorithms for specific forms of calibration, such as quantile or moment-based calibration.

**Audience:**

Yes

**Broader Impact Concerns:**

None.

**Claims And Evidence:**

No

**Requested Changes:**

## Experiments
- Add experiments to evaluate:
  - The specialized oracles derived in Appendix C.
  - Pathological scenarios with feedback loops, distribution shifts, adversarial environments to test the claimed robustness of the methods.
- Expand the evaluation of ORCA by:
  - Including additional baselines for comparison.
  - Testing on more datasets.

## Related Work
- Clarify the contributions relative to the current literature. Is the framework described entirely new? If not, what are the novel aspects compared to existing results and methods?

## Limitations
- Add a dedicated section discussing limitations of the proposed algorithms (ORCA and specialized oracles):
  - For example, ORCA is widely applicable, but its approximate optimization procedure lacks theoretical guarantees.
  - Are there any forms of calibration not supported by the framework or algorithms?
  - Discuss the computational feasibility of specialized algorithms. Are they practical for real-world applications?

## Minor Edits and Addtional Questions
- The paper reads very well. I suggest adding a paragraph at the end of the introduction to outline the structure of the paper, which would further improve readability.
- Conditions 1-3 appear very general. Could you provide examples (if any) of common scenarios where these assumptions fail?
- In the introduction, you state:
  > "We provide finite-sample calibration guarantees [...] to any form of calibration fitting into our framework."

  Are there commonly used forms of calibration that fall outside this framework? If so, these should be explicitly mentioned.

**Strengths And Weaknesses:**

## Strengths
 - _A very general framework._ The paper provides a unifying theoretical framework that encompasses a wide range of calibration measures, making it applicable to diverse forecasting tasks.
 - _Strong finite-sample guarantees._ The authors show that miscalibration decays at a rate of $O(1/\sqrt{T})$ under mild assumptions, even in nonstationary and adversarial environments.
 - _No-regret recalibration of existing models._ The framework extends to recalibrating forecasts from existing models, ensuring both calibration and no-regret performance while leveraging the inductive biases of these models.
 - _Clear and cohesive presentation_. The paper is well-structured and well-written. The framework and results are presented in a way that feels cohesive and easy to follow. The notation is clear.

## Weaknesses
The main weakness of this paper is the limited experimental evaluation.

While the authors propose specialized algorithms that solve the half space problem for specific calibration metrics, they provide no experiments that assess their computational feasibility and practical performance. The only experiments focus on the gradient-based method (ORCA). Since ORCA lacks theorethical guarantees, experimental validation is particularly relevant here. However, this method is tested on just one baseline and two datasets, limiting the scope of its evaluation.

The authors also claim their framework can “achieve general notions of calibration for nonstationary data caused by feedback loops, distribution shifts, time series, and adversarial environments.” While this is supported by their theoretical results, it remains unclear whether this can be achieved in practice using either the specialized algorithms or ORCA.

---

> ### Author Response · Authors · 2025-01-10
>
> Thank you for your feedback and suggestions! See our responses to each of your questions below.
>
> ## Experiments
>
> Based on your suggestions, we created a new experiment to evaluate ORCA on truly adversarial data. For this experiment, we consider a regression task with outcomes the unit interval $[0, 1]$. In each round, Forecaster predicts a distribution $p$ over the unit interval, then Nature chooses the outcome $y \in [0, 1]$ that maximizes the quantile calibration error (QCE) for that forecast. The goal of the forecaster is to minimize the QCE, so the task is a zero-sum game between Forecaster and Nature. The result of the experiment can be found at the following link:
>
> https://anonymous.4open.science/r/orca-adversarial/qce_adversarial.png
>
> The figure shows the mean QCE over 10 runs for each forecaster, with the shaded region indicating the standard deviation. This experiment also demonstrates how ORCA can be used as a base forecaster, as opposed to only a recalibration technique. Please feel free to follow up if you have any other questions about this experiment. We will add this experiment to the updated manuscript.
>
> Lastly, while experiments for the additional algorithms in the appendix are outside the scope of this work, we agree that a thorough empirical comparison is an interesting direction for future work.

---

> ### Author Response · Authors · 2025-01-10
>
> ## Related Work
> > Clarify the contributions relative to the current literature. Is the framework described entirely new? If not, what are the novel aspects compared to existing results and methods?
>
> The current literature uses Blackwell approachability to achieve calibration in online learning, focusing on the binary setting. The primary novel aspect of our framework is that we extend these ideas to a general class of payoffs, described by our Conditions 1-3. Our work can be viewed as extending the many forms of calibration used in the offline setting (quantile calibration, decision calibration, local calibration, etc.) to the online setting, using ideas inspired by the literature in the binary setting. Furthermore, our algorithmic approach is quite different from existing work, which previously focused on specialized algorithms for binary calibration. In contrast, ORCA is a plug-and-play algorithm which can easily be applied to new forms of calibration; a practitioner only needs to provide a differentiable payoff function. This also creates the new possibility to jointly enforce multiple forms of calibration. We will add this discussion to the literature review.

---

> ### Author Response · Authors · 2025-01-10
>
> ## Limitations
> > Add a dedicated section discussing limitations of the proposed algorithms (ORCA and specialized oracles):
>
> **Limitations**
> ORCA lacks the exact finite-sample guarantees from our framework due to the approximations used to facilitate optimization. Still, some approximate guarantees can be provided (see our response to Reviewer dQat). Additionally, while our theoretical results apply to nondifferentiable payoffs, in practice ORCA requires differentiability to facilitate gradient-based optimization. Smoothing applied to improve the behavior of gradient-based optimizers creates a deviation between the objective being optimized and the true calibration objective. In practice, we find that this distinction is small enough that performance is not impacted. Lastly, discontinuous or unbounded payoffs cannot be directly optimized. This is a fundamental limitation due to impossibility results. In practice, we apply clipping to handle unbounded payoffs (e.g., for no-regret recalibration with respect to negative log-likelihood) and kernel smoothing to handle discontinuities (e.g., for expected calibration error).
>
> For the specialized oracles described in the appendix, the central limitation is a lack of flexibility and difficulty in combining objectives. While ORCA can naturally balance multiple objectives by concatenating the payoff functions, the specialized oracles may produce conflicting forecasts for different objectives.
>
> We will add a subsection to Section 8 discussing these limitations.

---

> ### Author Response · Authors · 2025-01-10
>
> ## Minor Edits and Additional Questions
> > The paper reads very well. I suggest adding a paragraph at the end of the introduction to outline the structure of the paper, which would further improve readability.
>
> We’re glad you found that the paper reads well! Thanks for this suggestion. We will add a short outline to the end of the introduction to summarize the content of each section.
>
> > Conditions 1-3 appear very general. Could you provide examples (if any) of common scenarios where these assumptions fail?
>
> Yes these conditions are quite general! Here are some common scenarios where Condition 1 (Boundedness) and Condition 3 (Continuity) fail. Boundedness fails for no-regret recalibration with respect to the negative log-likelihood (NLL) scoring rule, since NLL is not bounded. Continuity fails for the popular Expected Calibration Error (ECE) payoff, due to the discontinuous binning of the ECE. These issues can be mitigated by adjusting the payoff, such as clipping the NLL to achieve boundedness or applying miniscule smoothing to the ECE to achieve continuity.
>
> > In the introduction you state “We provide finite-sample calibration guarantees [...] to any form of calibration fitting into our framework.” Are there commonly used forms of calibration that fall outside this framework? If so, these should be explicitly mentioned.
>
> Yes, the Expected Calibration Error falls outside our framework due to the discontinuities at the bin edges. We will add explicit mention of this, as well as how we handle the issue with miniscule kernel smoothing.

---

> ### Comment · Reviewer_XmLe · 2025-01-16
>
> Thank you for your clear and detailed responses to my questions about the related work, the limitations of the methods, and the additional points I raised.
>
> I appreciate the new experiment with adversarial data. While the scenarios involving feedback loops and distribution shifts remain unexplored, the adversarial experiments demonstrate ORCA’s strong worst-case robustness and align in spirit with the paper’s theoretical results for the oracle methods. I believe the new experimental section, with the added adversarial tests, provides sufficient evidence of the method’s relevance, and I look forward to reading it!
>
> Regarding the new experiments, how are the methods initialized? At each run, ORCA appears to start with a better QCE at step 0.
>
> Also, thank you for clarifying that experiments for the specialized oracles are outside the scope of this work. If these algorithms are not central to the proposed methodology, what makes them worth including? In which scenarios would they be useful, and are they practically relevant?

---

> > ### Author Response · Authors · 2025-01-20
> >
> > We appreciate your positive feedback on the new adversarial experiment and its relevance to our method!
> >
> > Regarding initialization, the methods are initialized identically with an empty history. Step 0 in the plot is the QCE after the first forecast, since we used a 0-indexing convention. To avoid confusion, we’ll adjust the plot to be 1-indexed. ORCA starts with a lower QCE because it produces a minimax-optimal initial forecast with respect to QCE, whereas the baselines adhere to the default behavior of their respective River library implementations (https://riverml.xyz/).
> >
> > Regarding the algorithms in the appendix, they were included to show that specialized methods with stronger guarantees can be derived. While these methods are not central to the proposed approach, they are relevant for scenarios like unmonitored systems, where convergence guarantees are valuable.

---

> > > ### Comment · Reviewer_XmLe · 2025-01-21
> > >
> > > Thank you for your response; this makes sense. Looking forward to reading the revised manuscript!

---

### Review · Reviewer_p6eq · 2025-01-02

**Summary Of Contributions:**

This paper proposes a novel framework of calibrated probabilistic forecasts for arbitrary sequence by formulating it as a zero sum game between the forecaster and the nature and leveraging the concept of Blackwell approachability from game theory.
Practical algorithms are proposed and numerical experiments results show the validity of their proposed algorithm.

**Audience:**

Yes

**Claims And Evidence:**

Yes

**Requested Changes:**

I have some questions:
1. In equation (2),  the halfspace is defined by the average payoff, then why it's $\bar{\pi}$ instead of $\bar{\pi}_T$?
2. “ For any discontinuous payoff, continuity can be achieved by smoothing the payoff on an arbitrarily small scale.” Can you further elaborate on this claim? For example, if the payoff is smoothed, then it is changed. Does this change have any influence?
3. The regret $R_T$ is defined with respect to the average performance, while in equation (7), $pi^{\text{REG}}(x,p,y)$ is not an averaged term. Is this a typo, otherwise how to understand it?
4. I kind of lost at the Recalibration and Theorem 5.2 of section 5.2. Though more details are provided in the appendix, I was wondering why it is placed in section 5.2. Moreover, it seems that recalibration is one of the main contribution of your paper, maybe more discussion in the main context would be better.
5. In the abstract, how to understand "accurate uncertainties" for the recalibrate method?

**Strengths And Weaknesses:**

This paper is in general clearly written. The perspective of formulating the forecast task as a zero-sum game played between the Forecaster and Nature and leveraging Blackwell approachability to achieve provable calibration is interesting. The algorithms motivated by the theoretical insights are practical and effective as proved by the experiments. I have some minor questions, please see the Requested Changes part for details.

---

> ### Author Response · Authors · 2025-01-10
>
> Thanks for your engagement and suggestions! We’re glad that you found our work interesting. See below for responses to each of your comments:
>
> > 1. In equation (2), the halfspace is defined by the average payoff, then why it’s $\overline{\pi}$ instead of $\overline{\pi}_T$?
>
> We use the notation $\overline{\pi}_T$ for the specific average payoff at time $T$, and $\overline{\pi}$ for an arbitrary average payoff (e.g., the infimum over $\overline{\pi} \in \mathcal{E}$ above equation (2)). We made the stylistic choice to omit the subscript in equation (2) to emphasize that the condition must hold for all average payoffs, not a single realized average payoff.
>
> > 2. “For any discontinuous payoff, continuity can be achieved by smoothing the payoff on an arbitrarily small scale.” Can you further elaborate on this claim? For example, if the payoff is smoothed, then it is changed. Does this change have any influence?
>
> Great question! Concretely, this smoothing can be implemented by convolving the payoff function with a kernel over the forecast. Applying this adjustment to the payoff does affect the corresponding notion of calibration, but it does not meaningfully change its interpretation.
>
> We can use binary calibration as an example. The standard (discontinuous) measure of miscalibration is given by the vector-valued payoff $\pi$, with entries given by $\pi_p(p_t, y_t) =  1_{\{ |p_t - p| \leq M/2 \}} \cdot (y_t - p_t)$, where $p = 1/M, …, (M - 1)/M$. Due to impossibility results for optimizing this discontinuous payoff online [1], we slightly smooth the indicator function, treating a forecast $p_t$ that is close to the discontinuity as partially in each neighboring bin. This preserves the spirit of the binary calibration metric, which is that, for forecasts close to  $p$, the frequency of $y=1$ should approach $p$. We will add a note about this discussion to the updated manuscript.
>
> [1] Dawid, A. Philip. "Comment: The impossibility of inductive inference." Journal of the American Statistical Association 80.390 (1985): 340-341.
>
> > 3. The regret $R_T$ is defined with respect to the average performance, while in equation (7), $\pi^\textrm{REG}(x, p, y)$ is not an averaged term. Is this a typo, otherwise how to understand it?
>
> Yes, this is a typo. The corrected expression is $\overline{\pi}_T^\textrm{REG}(x, p, y)$. Thank you for catching this!
>
> > 4. I kind of lost at the Recalibration and Theorem 5.2 of section 5.2. Though more details are provided in the appendix, I was wondering why it is placed in section 5.2. Moreover, it seems that recalibration is one of the main contribution of your paper, maybe more discussion in the main context would be better.
>
> Thanks for these suggestions! Section 5.2 is intended to provide a short introduction to the specialized algorithms we introduce in the appendix. The paragraph titled “Recalibration” in Section 5.2 is an application of these algorithms to no-regret recalibration, and can be viewed as a short addendum to our main discussion of recalibration in Section 4.2. We will refocus this paragraph to provide a more thorough description of the results in the appendix, and defer Theorem 5.2 to the appendix since it is not critical for the main discussion.
>
> > 5. In the abstract, how to understand "accurate uncertainties" for the recalibrate method?
>
> We use “accurate uncertainties” here as a synonym for calibrated uncertainties. To avoid confusion, we will rephrase this as “calibrated uncertainties”.

---

> ### Comment · Reviewer_p6eq · 2025-01-15
>
> Thanks for the responses, they address my questions.

---

### Author Response · Authors · 2025-01-10

We would like to thank the reviewers for their thoughtful feedback. We respond to each reviewer individually below, and we will incorporate these updates into the revised manuscript.

Based on the reviewer suggestions, we have also added an experiment with adversarial data. The experimental setup can be found in our response to reviewer XmLe, and the results can be found at the following link:

https://anonymous.4open.science/r/orca-adversarial/qce_adversarial.png

---

### Author Response · Authors · 2025-01-21
**Revised Manuscript Uploaded**

We have uploaded a revised manuscript incorporating the discussions below. We'd like to highlight the most important updates we made based on reviewer feedback:
- We added the new experiment with adversarial data in Figure 1 and Section 6.2.
- We added an analysis of the impacts of the approximation in Eq (9) to Section 5.1.
- We added a new discussion of limitations in Section 8, including how we handle settings that do not fit exactly into our framework.

We'd like to thank the reviewers again for their helpful suggestions which have strengthened our work!

---

### Decision · Action_Editor_b4GF · 2025-02-08

**Recommendation:** Accept as is

**Comment:**

The author's revision addressed concerns brought up during the reviewing process. In their revision, they:

- Added a new experiment with adversarial data to demonstrate the effects of their proposed method in specialized scenarios
- The authors discussed how the gradient-based relaxation of their method affects performance and computation.
- The authors improved their discussion of limitations.

Even before this revision, the paper was well-structured and clear, discussed theoretical and practical aspects of their algorithm, and brought a new perspective to online calibration. All reviewers agreed that this paper would be more compelling with additional experiments that demonstrated the proposed method (ORCA) in other environments and across more datasets; however the existing results provide evidence for the method.

As is, this paper meets the criteria for acceptance at TMLR.

**Audience:**

Calibration and online learning are important topics in machine learning. This paper focuses on these important problems and brings in a new game theoretic perspective. It will be of interest to the TMLR audience.

**Claims And Evidence:**

This paper focuses on finite-sample calibration guarantees for online learning. Their work is well founded in the existing statistical and game-theoretic results. Their proposed method is theoretically motivated, offers finite guarantees, and the authors provide a heuristic approximation for computational ease. The authors demonstrate their method on real-world experiments.

The evidence in this paper is high quality and clear. While more experimental evidence would strengthen the paper, the existing evidence more than meets the bar for TMLR.